# Task-Unaware Lifelong Robot Learning with Retrieval-based Weighted Local Adaptation

## Abstract

Real-world environments require robots to continuously acquire new skills while retaining previously learned abilities, all without the need for clearly defined task boundaries. Storing all past data to prevent forgetting is impractical due to storage and privacy concerns. To address this, we propose a method that efficiently restores a robot's proficiency in previously learned tasks over its lifespan. Using an Episodic Memory (EM), our approach enables experience replay during training and retrieval during testing for local fine-tuning, allowing rapid adaptation to previously encountered problems. Additionally, we introduce a selective weighting mechanism that emphasizes the most challenging segments of retrieved demonstrations, focusing local adaptation where it is most needed. This framework offers a scalable solution for lifelong learning without explicit task identifiers or implicit task boundaries, combining retrieval-based adaptation with selective weighting to enhance robot performance in open-ended scenarios.

## 1 Introduction

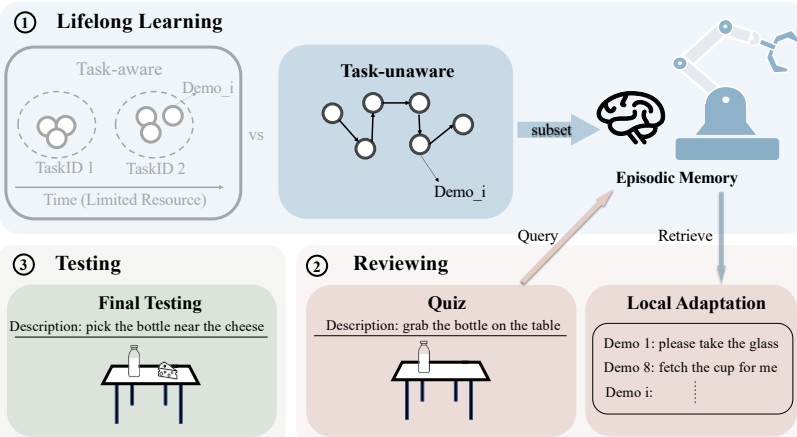

Figure 1: Method Overview. Our approach addresses the challenge of lifelong learning without distinct task boundaries. To emulate human learning patterns, we propose a method consisting of three phases: learning, reviewing, and testing. In the learning phase, the robot is exposed to various demonstrations, storing a subset of this data as episodic memory $\mathcal{M}$. During the reviewing phase, the method retrieves the most relevant data to fine-tune the policy network, enhancing performance in the final testing phase.

Lifelong learning seeks to endow neural networks with the ability to continually acquire new skills while retaining previously learned knowledge. This balance between stability and plasticity is crucial as models face sequences of tasks over time. While significant progress has been made in applying lifelong learning to domains such as computer vision (Huang et al., 2024; Du et al., 2024) and natural language processing (Shi et al., 2024; Razdaibiedina et al., 2023), the challenges are more pronounced in robotics. Robots are expected to adaptively learn and solve unseen tasks throughout

their operational lifespan (Thrun & Mitchell, 1995). Their interactions with dynamic environments introduce complexities absent in static data domains; a single misstep in task execution can result in complete failure. Moreover, robotics is constrained by limited data availability due to the expense and complexity of real-world interactions (Zhu et al., 2022; Du et al., 2023). These factors not only intensify the difficulty of continual learning in robotics but also demand more robust lifelong learning capabilities.

Lifelong robot learning typically requires robots to learn a sequence of tasks, each distinguished by domain, scenario, scene, or task goals (Liu et al., 2024; 2023). Existing lifelong robot learning algorithms often rely heavily on task IDs or boundaries. However, in dynamic real-world settings, it is impractical to predefine tasks or assign specific IDs, as robots are likely to encounter a vast array of unpredictable situations, with tasks that may be subdivided into smaller components of varying granularity. Therefore, approaches that depend on specific task identifications with clear boundaries are unrealistic and unscalable (Koh et al., 2021).

To address these challenges, we propose a novel task-unaware lifelong robot learning framework, which enables robots to continually learn and adapt without knowing task IDs or boundaries. We employ our method in manipulation scenarios based on the LIBERO benchmark (Liu et al., 2024). Our approach leverages pre-trained models to generate consistent embeddings across different tasks and training phases, thereby mitigating the embedding drift that often occurs in sequential learning scenarios (Liu et al., 2023; Kawaharazuka et al., 2024). We adopt Experience Replay (ER) baseline (de Masson D'Autume et al., 2019) to rehearse samples from previous tasks, helping to maintain learned skills and reduce forgetting.

Despite these measures, some degree of forgetting remains inevitable due to the multitasking nature of lifelong learning and the robot's limited access to previous demonstrations. Drawing inspiration from human learning processes — where individuals revisit knowledge they once knew but have forgotten details — we introduce an efficient local adaptation mechanism. Humans often perform quick reviews using limited resources, allowing them to efficiently regain proficiency without relearning all aspects of the task (Sara, 2000). Similarly, when the robot encounters a forgotten scenario, our mechanism enables it to swiftly adapt and regain skills through fast fine-tuning, using the same episodic memory employed for experience replay during training.

Given the indistinct task boundaries, we leverage retrieval-based mechanisms (Du et al., 2023; van Dijk et al., 2024; de Masson D'Autume et al., 2019) to retrieve data most similar to the testing scenario based on vision and language input similarities. To adapt the model effectively — especially focusing on the most challenging phases where the robot's performance deviates — we first perform a few episodes of rollouts to obtain "feedback" on the model's performance before local adaptation: these rollouts are then used for automatic selective weighting by comparing them with the retrieved demonstrations without human intervention (Spencer et al., 2022; Mandlekar et al., 2020). The weighted samples facilitate the local adaptation phase, thereby improving performance.

In summary, the key contributions of our solution are:

- **Task-unaware Retrieval-Based Local Adaptation**: During testing, the robot retrieves relevant past demonstrations from episodic memory to locally adapt the neural network, enabling it to quickly regain proficiency on previously encountered but forgotten scenarios without relying on explicit task IDs nor implicit task boundaries.

- **Selective Weighting Mechanism**: A weighting mechanism emphasizes the most challenging segments of the retrieved demonstrations, optimizing real-time adaptation.

- **Paradigm for Memory-Based Lifelong Robot Learning**: We demonstrate that our approach can be applied to different memory-based robotic lifelong learning algorithms during test time, serving as a paradigm for skill restoration.

## 2 RELATED WORKS

### 2.1 LIFELONG ROBOT LEARNING

Robots operating in continually changing environments need the ability to learn and adapt on-the-fly (Thrun, 1995). In recent years, lifelong robot learning has been applied to SLAM (Yin et al.,

2023; Gao et al., 2022; Vödisch et al., 2022), navigation (Kim et al., 2024), and task and motion planning scenarios (Mendez-Mendez et al., 2023). In the context of continual learning from demonstrations, robots can 1) acquire skills from non-technical users (Grollman & Jenkins, 2007), or 2) adapt to user-specific task preferences — Chen et al. (2023) proposed strategy mixture approach to efficiently model new incoming demonstrations, enhancing adaptability.

Furthermore, methods have been developed to improve manipulation capabilities over a robot's lifespan. Some approaches leverage previous data to facilitate forward transfer but suffer from catastrophic forgetting (Xie & Finn, 2022). Others maintain an expandable skill set to accommodate an increasing number of manipulation tasks (Parakh et al., 2024), or continually update models of manipulable objects for effective reuse (Lu et al., 2022). Large language models have been utilized to improve knowledge transfer (Bärmann et al., 2023; Tziafas & Kasaei, 2024; Wang et al., 2023), and hypernetworks with neural ODEs have been employed to remember long trajectories (Auddy et al., 2023) incrementally. Additionally, (Yang et al., 2022) evaluates how typical supervised lifelong learning methods can be applied in reinforcement learning scenarios for robotic tasks.

To standardize the investigation of lifelong decision-making and bridge research gaps, Liu et al. (2024) introduced LIBERO, a benchmarking platform for lifelong robot manipulation where robots learn multiple atomic manipulation tasks sequentially. Recent works exploring lifelong robot learning based on it include (Liu et al., 2023), which assigns a specific task identity to each task, and (Wan et al., 2024), which requires a pre-training phase to build an initial skill set before continual learning. However, catastrophic forgetting for lifelong robot learning remains an open challenge, especially when task IDs and boundaries are not available.

## 2.2 Task-unaware Continual Learning

Despite the success of continual learning with clearly labeled task sequences, there still remains a gap in progress within settings where algorithm is unaware of task boundaries both in training and in inference, an online situation more reflective of real-world scenarios. Many attempts (Lee et al., 2020; Chen et al., 2020; Ardywibowo et al., 2022) focus on learning specialized parameters using expanding network structures. Memory-based algorithms remain effective by prioritizing informative samples (Sun et al., 2022), removing less important training samples (Koh et al., 2021), improving decision boundaries (Shim et al., 2021), and increasing gradient diversity (Aljundi et al., 2019b). Methods aiming to exploit replay buffer (He et al., 2020; Mai et al., 2021; Caccia et al., 2021) have also demonstrated notable success. Moreover, researchers (Aljundi et al., 2019a) tackled the issue of implicit task boundaries in regularization-based lifelong learning methods (Kirkpatrick et al., 2017; Aljundi et al., 2018; Zenke et al., 2017) by introducing an online learning approach that consolidates knowledge upon detecting a loss "plateau". However, these approaches have not been explored in robotic applications that involve decision-making and physical interactions with the environment.

## 2.3 Information Retrieval for Robotics

Information retrieval techniques have been used to optimize robotic behaviors by retrieving relevant actions from memory in novel tasks (Du et al., 2023). For example, path following based on image retrieval improves visual navigation (van Dijk et al., 2024), and incremental learning helps humanoid robots adapt to new environments by recalling past behaviors (Bärmann et al., 2023). Retrieval has also enabled skill transfer from videos (Papagiannis et al., 2024) and affordance transfer for zero-shot manipulation (Kuang et al., 2024), allowing robots to manipulate objects without prior training.

## 2.4 Robot Learning with Adaptation

Recent advances have shown robots adapting to dynamic environments, such as executing agile flight in strong winds (O'Connell et al., 2022), adapting quadruped locomotion through test-time search (Peng et al., 2020), and generalizing manipulation skills from limited data (Julian et al., 2020). To enable few-shot or one-shot adaptation, meta-learning has been extensively explored (Finn et al., 2017a) and successfully applied to robotics (Kaushik et al., 2020; Nagabandi et al., 2018; Finn et al., 2017b). However, meta-learning methods typically assume access to a full distribution of tasks dur-

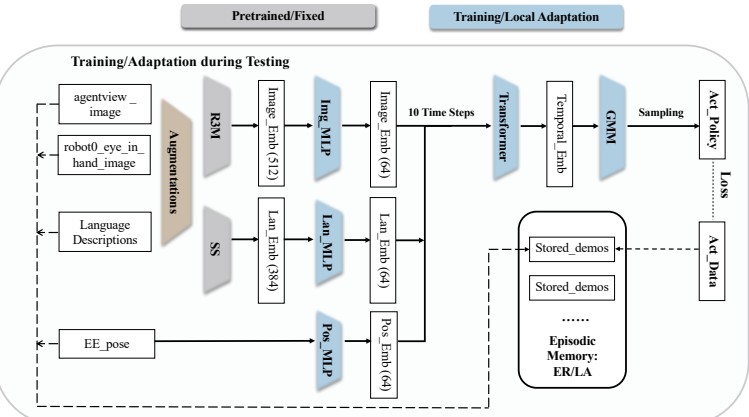

Figure 2: Policy Backbone Architecture used in Training and Testing. We input various data modalities into the system, including demonstration images, language descriptions, and the robot arm's proprioceptive input (joint and gripper states). Pretrained R3M (Nair et al., 2022) and (SentenceSimilarity, 2024) models process the image and language data respectively. Along with the proprioceptive states processed by an MLP, the embeddings are concatenated and passed through a Transformer to generate temporal embeddings. A GMM (Gaussian Mixture Model) is then used as the policy head to sample actions for the robot. Throughout both training and testing, we utilize episodic memory to store a subset of demonstrations gathered throughout the training process.

ing meta-training, with both training and testing performed on tasks sampled from this distribution. In contrast, our lifelong robot learning scenario operating sequentially lacks such access, presenting unique challenges of catastrophic forgetting.

## 3 PRELIMINARY

Unlike previous approaches (Liu et al., 2023; Kirkpatrick et al., 2017; Mallya & Lazebnik, 2018) that rely on explicit task identifiers or implicit task boundaries, such rigid categorization becomes impractical in open-ended environments. To model these realistic conditions, we define a set of tasks $\mathbb{T} = \{\mathcal{T}_k\}, k = 1, 2, \ldots, T$, where each task is represented by $\mathcal{T}_k$ encompassing various environmental settings (e.g., objects positions, robot initial states) and language descriptions (e.g., "pick the bottle and put it into basket", "Please place the bottle into basket"). From $\mathcal{T}_k$, we sample specific environmental settings and language descriptions to generate a concrete scenario $\mathcal{S}_n^k \sim p(\mathcal{T}_k)$, which serves as the basis for collecting demonstrations $\tau_n^k$. Multiple demonstrations form the training dataset $\mathcal{D}_k = \{\tau_n^k\}, n = 1, 2, \ldots, N$ for task $\mathcal{T}_k$.

Notably, multiple tasks may share overlapping distributions in either environmental settings or language descriptions, and our lifelong learning approach operates without access to task identifiers or boundaries between different tasks $\mathcal{T}_k$. This natural setting closely mirrors real-world conditions, where it is difficult to determine which task generated a given scenario. This ambiguity underpins the proposed method's task-unaware design: instead of relying on task identification, it focuses on retrieving relevant information.

Our robot utilizes a visuomotor policy learned through behavior cloning to execute manipulation tasks by mapping sensory inputs and task descriptions to motor actions. The policy is trained by minimizing the discrepancy between the predicted actions and the expert actions derived from demonstrations. Specifically, we optimize the following loss function across a sequence of tasks $\mathbb{T}$ with $\mathcal{D}_k$. Notably, $\mathcal{D}_k$ is not fully accessible for $k < K$ due to the use of experience replay from Episodic Memory $\mathcal{M}$, where $K$ denotes the current task:

$$\theta^* = \arg\min_\theta \frac{1}{K} \sum_{k=1}^{K} \mathbb{E}_{(o_t, a_t) \sim \mathcal{D}_k, \, g \sim G_k} \left[ \sum_{t=0}^{l_k} \mathcal{L}\left(\pi_\theta(o_{\leq t}, g), \, a_t\right) \right], \tag{1}$$

where $\theta$ denotes the model parameters, $l_k$ represents the number of samples for task $k$, $o_{\leq t}$ denotes the sequence of observations up to time $t$ in demonstration $n$ (i.e., $o_{\leq t} = (o_0, o_1, \ldots, o_t)$), and $a_t$ is the expert action at time $t$. The set $G_k$ comprises various goal descriptions for task $\mathcal{T}_k$, with $g$ being a sampled goal description from $G_k$. The policy output, $\pi_\theta(o_{\leq t}, g)$, is conditioned on both the observation sequence and the goal description.

By optimizing this objective function, the policy effectively continues learning new tasks and skills in its life span, without the need for explicit task labels, thereby facilitating robust and adaptable task-unaware continual learning.

# 4    RETRIEVAL-BASED WEIGHTED LOCAL ADAPTATION FOR LIFELONG ROBOT LEARNING

In this section, we outline our proposed method depicted in Figure 1, with corresponding pseudocode in Algorithm 1. To effectively interact with complex physical environments, the network integrates multiple input modalities, including visual inputs from workspace and wrist cameras, proprioceptive inputs of joint and gripper states, and task descriptions.

Instead of training all modules jointly in an end-to-end manner, we employ pretrained visual and language encoders that leverage prior semantic knowledge. Pretrained encoders enhance performance on downstream manipulation tasks (Liu et al., 2023) and are well-suited to differentiate between various scenarios and tasks without relying on explicit task identifiers or clear task boundaries. Their consistent representations when new tasks continue to come is essential for managing multitask problems and retrieving relevant data to support our proposed local adaptation during test time.

When learning new tasks, the robot preserves previously acquired skills by replaying prior manipulation demonstrations stored in an episodic memory $\mathcal{M}$, which contains a small subset of previous task demonstrations (Chaudhry et al., 2019). Trained with the combined data from the latest scenarios and episodic memory $\mathcal{M}$, the model can acquire new skills while mitigating catastrophic forgetting of old tasks, thereby maintaining a balance between stability and plasticity (Wang et al., 2024). Figure 2 illustrates the network architecture, and implementation details are provided in Section A.2.

## 4.1    DATA RETRIEVAL

During deployment, we first retrieve the most relevant demonstrations from episodic memory $\mathcal{M}$ based on similarity to the current scenario. Due to the unclear task boundaries, some tasks share similar visual observations but differ in their task objectives, while others have similar goals but involve different backgrounds, objects, etc. To account for these variations, we compare both visual inputs from the workspace camera (Du et al., 2023) and task descriptions (de Masson D'Autume et al., 2019) using $L_2$ distances of their embeddings. The retrieval process follows a simple rule:

$$\mathcal{D}_R = \alpha_v \cdot \mathcal{D}_v + \alpha_l \cdot \mathcal{D}_l, \tag{2}$$

where $\mathcal{D}_R$ is the weighted retrieval distance, $\mathcal{D}_v$ represents the distance between the embeddings of the scene observation from the workspace camera, and $\mathcal{D}_l$ depicts the distance between the task description embeddings. The parameters $\alpha_v$ and $\alpha_l$ control the relative importance of visual and language-based distances. Based on the distances $\mathcal{D}_R$, the most relevant demonstrations can be retrieved from $\mathcal{M}$, as illustrated from Figure 8.

## 4.2    WEIGHTED LOCAL ADAPTATION

### 4.2.1    LEARN FROM ERRORS BY SELECTIVE WEIGHTING

To make the best use of the limited data, we enhance their utility by assigning weights to critical or vulnerable segments in each retrieved demonstration. Specifically, before testing, the robot performs several rollouts on the encountered task using the existing model trained during the lifelong learning phase. This procedure allows us to evaluate the model's performance and identify any forgetting effects, akin to a preliminary quiz before the final exam (as illustrated in step 2, the *reviewing phase* in Figure 1).

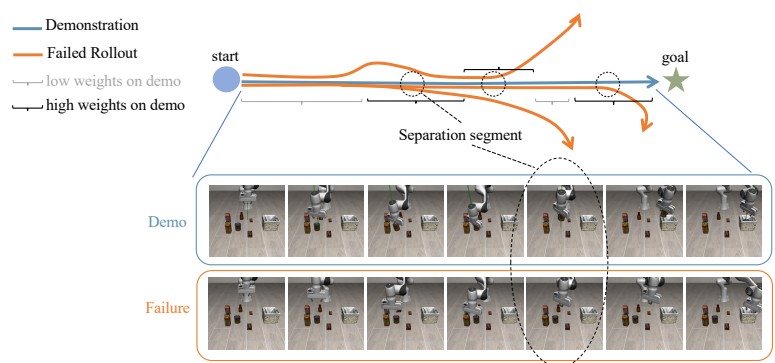

Figure 3: Trajectory and Weighting Visualizations. To identify the point of failure, we compute the similarity between the retrieved demonstrations and failed trajectories at each frame. Once the separation segment is detected, higher weights are assigned to the frames in the segment of retrieved demonstrations during local adaptation.

When failed trajectories are identified, we compare each image in the retrieved demonstrations against all images from the failed trajectories using the $L_2$ distances of their embeddings. This comparison yields a distance vector for each demonstration, where each value represents the minimal distance between a demonstration frame and all images from the failed rollouts. This metric determines whether a particular frame has occurred during the rollout. Through this process, we identify the **Separation Segment** — frames in the demonstrations where the behavior deviates from what was executed during the failed rollouts (see Figure 3). Since these Separation Segments highlight behaviors that should have occurred but did not, we consider them vulnerable segments that contribute to the failure. We assign higher weights to these frames which will scale the losses during local adaptation. Detailed heuristics and implementation specifics are provided in Appendix A.4.

### 4.2.2 ADAPTATION WITH FAST FINETUNING

Finally, we fine-tune the network's parameters to better adapt to the current task using the retrieved demonstrations, focusing more on the difficult steps identified through selective weighting. Notably, the episodic memory $\mathcal{M}$ contains the same data used during training for experience replay and during deployment for local adaptation. No additional demonstrations are available to the robot at test time. Despite this limited data, our experiments demonstrate that the model can effectively recover learned skills and improve its performance across various tasks. Overall, the proposed weighted adaptation is formalized as follows:

$$\theta^* = \arg\min_{\theta} \sum_{n=1}^{\tilde{N}} \sum_{t=1}^{l_n} w_{t,n} \mathcal{L}\left(\pi_\theta(o_{\leq t,n}, g_n), a_{t,n}\right) \tag{3}$$

where $\tilde{N}$ is the number of retrieved demonstrations, $l_n$ is the length of demonstration $n$, and $w_{t,n}$ is the weight assigned to sample $t$ in demonstration $n$. The variables $o_{\leq t,n}$ and $a_{t,n}$ denote the sequence of observations up to time $t$ and the corresponding expert action, respectively, while $g_n$ is the goal description for demonstration $n$. The parameter $\theta$ represents the network's parameters before adaptation.

## 5 EXPERIMENTS

We conduct a comprehensive set of experiments to evaluate the effectiveness of our proposed retrieval-based weighted local adaptation method for lifelong robot learning. Specifically, our experiments aim to address the following key questions:

1. **Effect of Blurry Task Boundaries:** How do blurry task boundaries influence the model's performance and data retrieval during testing?

2. **Advantages of Retrieval-Based Adaptation:** Does retrieval-based weighted local adaptation enhance the robot's performance across diverse tasks?

3. **Impact of Selective Weighting:** Is selective weighting based on rollout errors effective in improving task performance?

4. **Generalizability:** Can our method be applied to different memory-based lifelong robot learning approaches, serving as a paradigm that enhances the performance during test time by restoring previous knowledge and skills?

5. **Robustness:** Due to blurry task boundaries and retrieval imprecision, the retrieved demonstrations may not necessarily belong to the same task. How resilient is our method to inaccuracies in memory retrieval?

## 5.1 EXPERIMENTAL SETUP

### 5.1.1 BENCHMARKS

We evaluate our proposed methods using LIBERO (Liu et al., 2024): `libero_spatial`, `libero_object`, `libero_goal`, and `libero_different_scenes`. These environments feature a variety of objects and layouts. The first three benchmarks all include 10 distinct tasks (e.g., Put the bottle into the basket. Open the middle drawer of the cabinet.), each with up to 50 demonstrations collected in simulation with different initial states of objects and the robot. Specifically, `libero_different_scenes` is created from LIBERO's provided `LIBERO_90`, which encompasses 20 tasks from distinct scenes.

For each task, we paraphrased the assigned single goal description into diverse descriptions to obscure task boundaries (See Figure 6). These enriched descriptions were generated by rephrasing the original task descriptions from the benchmark using a large language model provided by *Phi-3-mini-4k-instruct* Model (mini-4k instruct, 2024), ensuring consistent meanings while varying phraseology and syntax. Please see Section A.3 for more details.

### 5.1.2 BASELINES

We evaluate our proposed method against the following baseline approaches:

1. **Elastic Weight Consolidation (EWC)** (Kirkpatrick et al., 2017): A regularization-based approach that constrains updates to the network's parameters to prevent catastrophic forgetting of previously learned tasks.

2. **Experience Replay (ER)** (Chaudhry et al., 2019): A core component of our training setup, ER utilizes stored episodic memory to replay past demonstrations, helping the model maintain previously acquired skills and mitigate forgetting. As a baseline, we evaluate the standalone performance of ER without additional retrieval-based adaptation techniques.

3. **Average Gradient Episodic Memory (AGEM)** (Hu et al., 2020): Employs a memory buffer to constrain gradients during the training of new tasks, ensuring that updates do not interfere with performance on earlier tasks.

4. **AGEM with Weighted Local Adaptation (AGEM-WLA):** An extension of AGEM that incorporates weighted local adaptation during the testing phase, enhancing the model's ability to adapt to specific tasks based on retrieved demonstrations. This allows us to assess the generalizability of our proposed method as a paradigm framework on other memory-based lifelong learning approaches.

5. **PackNet** (Mallya & Lazebnik, 2018): An architecture-based lifelong learning algorithm that iteratively prunes the network after training each task, preserving essential nodes while removing less critical connections to accommodate subsequent tasks. However, its pruning and post-training phases rely heavily on clearly defined task boundaries, making PackNet a reference baseline when task boundaries are well-defined.

### 5.1.3 METRICS

Our primary focus is on the success rate of task execution, as it is a crucial metric for manipulation tasks in interactive robotics. Consequently, we adopt the **Average Success Rate (ASR)** as our primary evaluation metric to address the challenge of catastrophic forgetting within the lifelong learning framework, evaluating success rates on three random seeds across all diverse tasks within

the same benchmark. Noteworthy, our results in both Table 1 and 2 are computed based on all tasks within each benchmark. Therefore, the differing levels of task difficulties and different success rates across tasks contribute to the high variance observed in the reported results.

### 5.1.4 MODEL, TRAINING, AND EVALUATION

As illustrated in Figure. 2, our model utilizes pretrained encoders for visual and language inputs: R3M (Nair et al., 2022) for visual encoding, Sentence Similarity model (SS Model) (SentenceSimilarity, 2024) for language embeddings, and a trainable MLP-based network to encode proprioceptive inputs. Embeddings from ten consecutive time steps are processed through a transformer-based temporal encoder, with the resulting output passed to a GMM-based policy head for action sampling. Specifically, R3M, a ResNet-based model trained on egocentric videos using contrastive learning, captures temporal dynamics and semantic features from scenes, while Sentence Similarity Model captures semantic relationships in task descriptions, enabling the model to differentiate between various natural language instructions.

The model first undergoes a lifelong learning phase, where it is trained sequentially on $10$ or $20$ tasks, depending on the specific benchmark, with each task trained for $50$ epochs. After training on each task, only random $20\%$ of the demonstrations from that task are stored in the episodic memory $\mathcal{M}$, which is used for experience replay to maintain learned knowledge. Every $10$ epochs, we check the model's performance and save the version that achieves the highest Success Rate to prevent over-fitting.

After training on all tasks sequentially, we conduct *reviewing* and *testing* on various scenarios sampled from each task for comprehensive analysis. During the *reviewing* stage, we firstly evaluate potential forgetting by having the agent perform $10$ rollout episodes on the deployment scenario $\mathcal{S}_{deploy}$—referred to as a *quiz* phase. We then retrieve the most similar demonstrations from $\mathcal{M}$ and fine-tune the model for only $20$ epochs using the retrieved demonstrations with selective weighting. Finally, we evaluate the adapted model for $20$ episodes—the *final testing* phase—to assess performance improvements. All training, local adaptation, and testing in the benchmarks are conducted using three random seeds ($1$, $21$, and $42$) to reduce the impact of randomness.

## 5.2 RESULTS

### 5.2.1 COMPARISON WITH BASELINES

To address Question 2, we compared our proposed method, Weighted Local Adaptation (ER-WLA), with all baseline approaches. As shown in Table 1, ER-WLA consistently outperforms baselines of EWC, AGEM, ER, and AGEM-WLA, which do not rely on explicit task IDs. By incorporating local adaptation during test time — our method mirrors how humans review and reinforce knowledge when it is partially forgotten — the continually learning robot could also regain its proficiency on previous tasks.

In contrast, PackNet serves as a reference method, as it requires well-defined task boundaries. However, as the number of tasks increases, the network's trainable capacity under PackNet diminishes, leaving less flexibility for future tasks. This limitation becomes evident in the `libero_different_scenes` benchmark, which includes $20$ tasks. PackNet's success rate drops significantly for later tasks, resulting in poor overall performance and highlighting its constraints on plasticity compared with our proposed ER-WLA approach.

Additionally, when we applied WLA to the AGEM baseline (resulting in AGEM-WLA), it also improved its performance, demonstrating the effectiveness of our method as a paradigm for memory-based lifelong robot learning methods. These findings also support our conclusions regarding Question 4.

### 5.2.2 ABLATION STUDIES

We performed two ablation studies to validate the effectiveness of our implementation choices and address Questions 1, 3, and 5.

Table 1: Comparison with Baselines, the average success rates and standard deviations across various baselines are shown below. We provide PackNet's performance on the right as a reference point for cases where task boundaries are accessible. Both EWC and vanilla AGEM demonstrate weak performance across all benchmarks, while ER performs better due to memory replay. Under our weighted local adaptation (WLA) paradigm, the WLA-enhanced versions of ER and AGEM show significant improvements over their vanilla counterparts, highlighting the effectiveness of WLA.

| Method | EWC | AGEM | AGEM-WLA | ER | ER-WLA | PackNet |
|---|---|---|---|---|---|---|
| *libero_spatial* | $0.0 \pm 0.0$ | $7.33 \pm 14.25$ | $35.83 \pm 15.71$ | $15.67 \pm 13.50$ | $\mathbf{39.83 \pm 19.85}$ | $53.17 \pm 24.72$ |
| *libero_object* | $1.50 \pm 3.26$ | $27.17 \pm 22.77$ | $51.17 \pm 24.13$ | $56.50 \pm 19.88$ | $\mathbf{62.33 \pm 18.69}$ | $73.77 \pm 16.97$ |
| *libero_goal* | $0.33 \pm 1.83$ | $10.83 \pm 16.03$ | $58.67 \pm 25.93$ | $52.33 \pm 22.16$ | $\mathbf{62.33 \pm 28.75}$ | $66.33 \pm 24.88$ |
| *libero_different_scenes* | $2.58 \pm 8.98$ | $20.43 \pm 25.55$ | $41.75 \pm 33.50$ | $34.08 \pm 28.55$ | $\mathbf{45.17 \pm 31.86}$ | $32.92 \pm 44.03$ |

**Selective Weighting.** In the first ablation, we evaluated the impact of selective weighting on `libero_spatial`, `libero_object`, and `libero_goal` benchmarks to demonstrate its importance for effective local adaptation. We compared two variants of our method: 1) **ER-ULA**, which applies uniform local adaptation without selective weighting, adapting retrieved demonstrations uniformly; 2) **ER-WLA**, which incorporates selective weighting during test-time adaptation. Both methods are trained with experience replay.

Since early stopping during local adaptation at test time is infeasible, and training can be unstable, particularly regarding manipulation success rates, we conducted adaptation using three different numbers of epochs — 15, 20, and 25 — followed by final testing. The results, presented in Table 2, indicate that selective weighting enhances performance across different adaptation durations and various benchmarks, confirming our hypothesis in Question 3.

**Language Encoding Model.** To investigate the impact of language encoders under blurred task boundaries with paraphrased descriptions, we ablated the choice of language encoding model. Specifically, we compared our chosen Sentence Similarity (SS) Model, which excels at clustering semantically similar language descriptions, with BERT, the default language encoder from LIBERO. We selected the `libero_goal` benchmark for this study because its tasks are visually similar, making effective language embedding crucial for distinguishing tasks and aiding data retrieval for local adaptation.

Our experimental results yield the following observations:

(1) As illustrated in Figure 4 (a) and (b), the PCA results show that the SS Model effectively differentiates tasks, whereas BERT struggles, leading to inadequate task distinction. Consequently, as shown in Figure 4 (c), the model trained with BERT embeddings on `libero_goal` performs worse than the one trained with SS Model embeddings.

(2) Due to this limitation, BERT is unable to retrieve the most relevant demonstrations (those most similar to the current task from the episodic memory $\mathcal{M}$). As a result, Retrieval-based WLA with BERT does not achieve optimal performance. These two findings address Question 1.

(3) Interestingly, from Figure 4 (c), despite BERT's low Retrieval Accuracy (RA), if it attains a moderately acceptable rate (e.g., 0.375), the local adaptation using data retrieved based on BERT embeddings can still enhance model performance during test time. This demonstrates the robustness and fault tolerance of our proposed approach, further addressing Question 4 and 5.

## 6 CONCLUSION AND DISCUSSION

In this paper, we introduced a novel task-unaware lifelong robot learning framework that combines retrieval-based local adaptation with selective weighting during test time. Our approach enables robots to continuously learn and adapt in dynamic environments without explicit task identifiers or predefined boundaries. Leveraging an episodic memory $\mathcal{M}$, our method retrieves relevant past demonstrations based on visual and language similarities, allowing the robot to fine-tune its policy locally. The selective weighting mechanism enhances adaptation by prioritizing the most challenging segments of the retrieved demonstrations. Notably, our framework is not only robust, but is

Table 2: Ablation Study on Selective Weighting. This table presents the performance of success rates with uniform (ULA) and weighted (WLA) local adaptation across 15, 20, and 25 epochs of adaptation under three random seeds, with evaluations conducted on all 10 tasks within the benchmarks: *libero_spatial*, *libero_object*, and *libero_goal*. Compared to ULA, the weighted scheme improves the method's performance on most benchmarks.

| Benchmark | Method | 15 Epochs | 20 Epochs | 25 Epochs | Overall ASR (%) |
|---|---|---|---|---|---|
| | | ASR (%) | ASR (%) | ASR (%) | |
| *libero_spatial* | ER-ULA | $35.33 \pm 21.21$ | $38.17 \pm 14.76$ | $\mathbf{38.16 \pm 17.19}$ | $37.22 \pm 17.77$ |
| | ER-WLA | $\mathbf{36.16 \pm 18.55}$ | $\mathbf{39.83 \pm 19.84}$ | $37.83 \pm 17.70$ | $\mathbf{37.94 \pm 18.57}$ |
| *libero_object* | ER-ULA | $57.83 \pm 25.14$ | $60.67 \pm 22.96$ | $58.00 \pm 21.84$ | $58.83 \pm 23.13$ |
| | ER-WLA | $\mathbf{58.00 \pm 22.35}$ | $\mathbf{62.33 \pm 18.70}$ | $\mathbf{61.50 \pm 24.36}$ | $\mathbf{60.61 \pm 21.76}$ |
| *libero_goal* | ER-ULA | $61.33 \pm 28.43$ | $62.00 \pm 29.61$ | $66.17 \pm 27.22$ | $63.17 \pm 28.20$ |
| | ER-WLA | $\mathbf{62.83 \pm 28.15}$ | $\mathbf{62.33 \pm 28.76}$ | $\mathbf{67.50 \pm 28.82}$ | $\mathbf{64.22 \pm 28.35}$ |

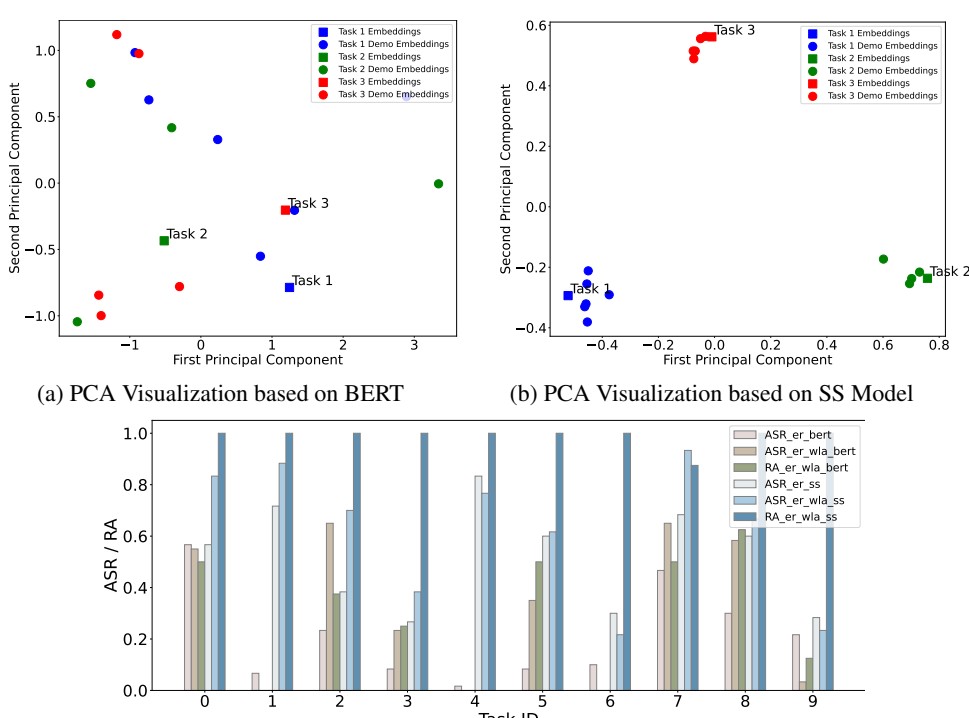

(a) PCA Visualization based on BERT  (b) PCA Visualization based on SS Model

(c) Bar Chart of Average Success Rates and Retrieval Accuracy across 10 tasks

Figure 4: In Figure 7a and Figure 7b, Principal Component Analysis (PCA) is used to visualize the distribution of language embeddings of 3 tasks from BERT and Sentence Similarity (SS), respectively. In Figure 4c, SS model, which distinguishes task descriptions, has higher success rate and retrieval accuracy than BERT.

compatible with various memory-based lifelong learning methods, enhancing a robot's ability to perform previously learned tasks as a paradigm.

A key challenge lies in the selective weighting process, particularly in finding the Separation Segment. Real-world noise, the multimodal nature of manipulation actions, and varying semantic information make it sometimes difficult to accurately identify Separation Segment in demonstration trajectories. Addressing this issue will be the focus of our future work to further improve our approach.

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

# A  APPENDIX

## A.1  NOTATIONS

Table 3: Mathematical Notations

| Symbol | Description |
|---|---|
| $k$ | Index of tasks, $k = 1, \ldots, K$ |
| $K$ | Total number of tasks |
| $n$ | Index of retrieved demonstrations |
| $\tilde{N}$ | Number of retrieved demonstrations |
| $i$ | Index of samples within a demonstration |
| $t$ | Time step |
| $l_k$ | Number of samples for task $k$ |
| $l_n$ | Length of retrieved demonstration $n$ |
| $\mathcal{T}_k$ | Task $k$ (represented by multiple goal descriptions) |
| $\mathcal{D}_k$ | Set of demonstrations for task $k$ |
| $\tau_i^k$ | Demonstration (trajectory) $i$ for task $k$ |
| $\mathcal{M}$ | Episodic memory buffer |
| $\mathbf{o}_t$ | Observation vector at time $t$ |
| $\mathbf{o}_{\leq t}$ | Sequence of observation vectors up to time $t$ to deal with partial observability |
| $\mathbf{a}_t$ | Action vector at time $t$ |
| $\mathbf{a}_t^k$ | Action vector at time $t$ for task $k$ |
| $x_{i,n}$ | Input of sample $i$ in retrieved demonstration $n$ |
| $y_{i,n}$ | Label (action) of sample $i$ in retrieved demonstration $n$ |
| $\theta$ | Model parameters |
| $\theta^*$ | Optimal model parameters |
| $\theta_k$ | Model parameters after adaptation on task $k$ |
| $\pi_\theta$ | Policy parameterized by $\theta$ |
| $\pi_\theta(\mathbf{s}_{\leq t}, \mathcal{T}_k)$ | Policy output given states up to time $t$ and task $\mathcal{T}_k$ |
| $\mathcal{L}$ | Loss function |
| $p(y \mid x; \theta)$ | Probability of label $y$ given input $x$ and parameters $\theta$ |
| $w_{i,n}$ | Weight assigned to sample $i$ in retrieved demonstration $n$ during adaptation |
| $\mathbb{E}$ | Expectation operator |
| $g_i$ | Goal descriptions in task $\mathcal{T}_k$ |

## A.2  IMPLEMENTATION AND TRAINING DETAILS

### A.2.1  NETWORK ARCHITECTURE AND MODULARITIES

Table 4 summarizes the core components of our network architecture, while Table 5 details the input and output dimensions.

### A.2.2  TRAINING HYPERPARAMETERS

Table 6 provides a summary of the essential hyperparameters used during training and local adaptation. The model was trained on a combination of **A40**, **A100**, and **L40S** GPUs, while we also leveraged multi-GPU configurations to accelerate the training process. For each task, demonstration data was initially collected and provided by LIBERO benchmark. However, due to version discrepancies that introduced visual and physical variations in the simulation, we reran the demonstrations with the latest version to obtain updated observations. It is important to note that occasional rollout failures occurred because different versions of RoboMimic Simulation (Mandlekar et al., 2021) utilize varying versions of the MuJoCo Engine (Todorov et al., 2012).

Task performance was evaluated every 10 epochs using 20 parallel processes to maximize efficiency. The best-performing model from these evaluations was retained for subsequent tasks. After training on each task, we reassessed the model's performance across all previously encountered tasks.

Table 4: Network architecture of the proposed Model.

| Module | Configuration |
|---|---|
| Pretrained Image Encoder | ResNet-based R3M (Nair et al., 2022), output size: 512 |
| Image Embedding Layer | MLP, input size: 512, output size: 64 |
| Pretrained Language Encoder | Sentence Similarity (SS) Model (SentenceSimilarity, 2024), output size: 384 |
| Language Embedding Layer | MLP, input size: 384, output size: 64 |
| Extra Modality Encoder (Proprio) | MLP, input size: 9, output size: 64 |
| Temporal Position Encoding | sinusoidal positional encoding, input size: 64 |
| Temporal Transformer | heads: 6, sequence length: 10, dropout: 0.1, head output size: 64 |
| Policy Head (GMM) | modes: 5, input size: 64, output size: 7 |

Table 5: Inputs and Output Shape.

| Modularities | Shape |
|---|---|
| Image from Workspace Camera | $128 \times 128 \times 3$ |
| Image from Wrist Camera | $128 \times 128 \times 3$ |
| Max Word Length | 75 |
| Joint States | 7 |
| Gripper States | 2 |
| Action | 7 |

### A.2.3 BASELINE DETAILS

We follow the implementation of all baselines and hyperparameters for individual algorithms from (Liu et al., 2024), maintaining the same backbone model and episodic memory structure as in our approach. During the training phase, we also apply the same learning hyperparameters outlined in Table 6.

### A.3 DETAILS ABOUT TASK-UNAWARE SETTING

In this paper, we blur task boundaries by using multiple paraphrased descriptions that define the task goals. The following section elaborate more details about our dataset and process of task description paraphrase.

### A.3.1 DATASETS STRUCTURE

Our dataset inherent the dataset from LIBERO (Liu et al., 2024), maintaining all the attributes and data. Additionally, we add *demo description* to each demonstration to achieve task unawareness and *augmented description* to augment language description during training (See Figure 5). Unlike the dataset from LIBERO, which groups demonstrations together under one specific task, our dataset wrap all demonstrations with random order to eliminate the task boundary.

### A.3.2 DESCRIPTION PARAPHRASE

We leverage the Phi-3-mini-4k-instruct model (mini-4k instruct, 2024) to paraphrase the task description. The process and prompts that we use are illustrated in Figure 6. As shown for the `libero_spatial` task in Figure 7, both BERT and Sentence Similarity Model struggle to distinguish tasks based on embeddings from the paraphrased descriptions. This observation further underscores the task-blurry setting in our experiments.

---

[1]For each task, demonstration data was collected from LIBERO, but due to differences in simulation versions, the demonstrations were rerun in the current simulation to collect new observations, with the possibility of occasional failures during rollout (see Subsection A.2.2 for details).

Table 6: Hyperparameter for Training and Adaptation.

| Hyperparameter | Value |
|---|---|
| Batch Size | 32 |
| Learning Rate | 0.0001 |
| Optimizer | AdamW |
| Betas | $[0.9, 0.999]$ |
| Weight Decay | 0.0001 |
| Gradient Clipping | 100 |
| Loss Scaling | 1.0 |
| Training Epochs | 50 |
| Image Augmentation | Translation, Color Jitter |
| Evaluation Frequency | Every 10 epochs |
| Number of Demos per Task | Up to 50 [1] |
| Number of Demos per Task in $\mathcal{M}$ ($\tilde{N}$) | 8 |
| Rollout Episodes before Adaptation | 10 |
| Distance weights $[\alpha_v, \alpha_l]$ for *libero_spatial* and *libero_object* | $[1.0, 0.5]$ |
| Distance weights $[\alpha_v, \alpha_l]$ for *libero_goal* | $[0.5, 1.0]$ |
| Distance weights $[\alpha_v, \alpha_l]$ for *libero_different_scenes* | $[1.0, 0.1]$ |
| Weights Added for Separation Segments | 0.3 |
| Clipping Range for Selective Weighting | 2 |
| Default Local Adaptation Epochs | 20 |

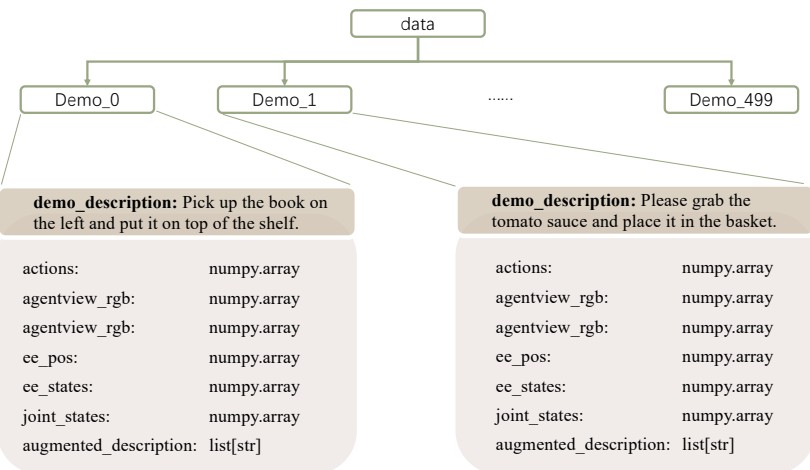

Figure 5: Data Structure

## A.4 DETAILS ABOUT SELECTIVE WEIGHTING

In this section, we introduce our Selective Weighting mechanism in detail.

### A.4.1 DETAILED HEURISTICS AND IMPLEMENTATIONS

To assign weights to retrieved demonstrations, we analyze the image embedding distance between demonstration and failed rollout trajectories. Typically, the embedding distance increases as the failed rollout diverges from the demonstration. We selectively add weights for the frames in the retrieved demonstration using the **Embedding Distance Curve (EDC)**, derived from the **Embedding Distance Matrix (EDM)**, as illustrated in Figure 9.

Due to the multi-modal nature of robotic actions and visual observation noise, raw embedding distances can be erratic. To mitigate this, we smooth the **EDC** using a moving average window. Despite smoothing, the curve may remain jittery, making it difficult to pinpoint a single divergence point.

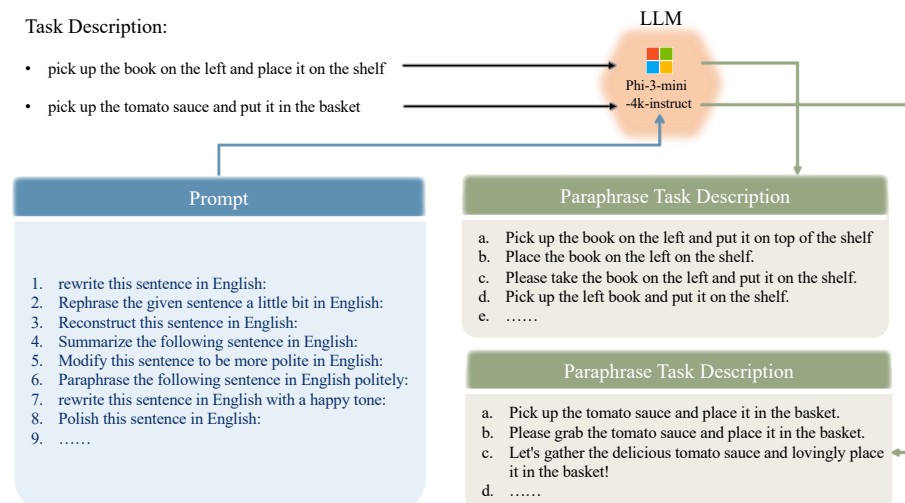

Figure 6: Paraphrase Description

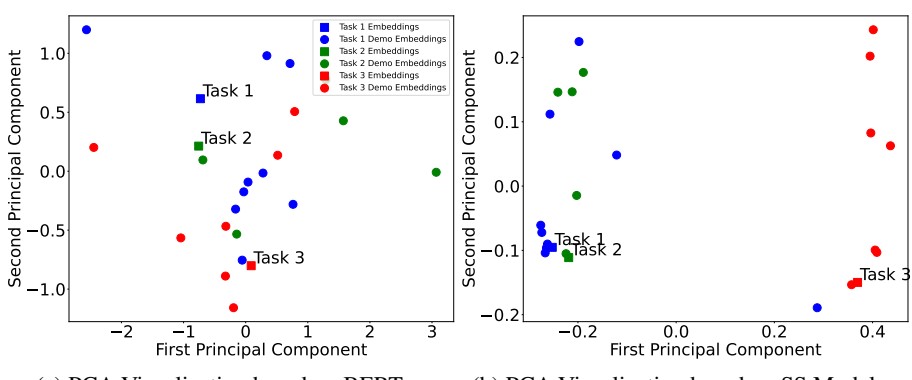

(a) PCA Visualization based on BERT      (b) PCA Visualization based on SS Model

Figure 7: Task Blurry Effect on `libero_spatial` benchmark. After paraphrasing the task descriptions, both Bert and SS models struggle to distinguish the tasks in `libero_spatial`.

Therefore, we identify a range of frames, termed the **Separation Segment**, where the distances increase, indicating vulnerable steps that lead to task failure.

We apply two thresholds to identify the segment: a lower threshold at $\frac{1}{8}$ and an upper threshold at $\frac{1}{3}$ of the maximum observed distance in EDC. We locate frames where the smoothed embedding distance falls within this range, focusing on the last occurrence to account for initial divergences that may later converge. We then extend this segment by $15$ frames before and after to mitigate noise effects.

For each frame within the Separation Segment, we increment the corresponding weight in the initially uniform weight vector by $0.3$. This process is repeated for up to five failed rollouts per retrieved demonstration. After processing all demonstrations, we clip the weights to a maximum of $2$ and normalize the weight vector to maintain consistent loss scaling and ensure stable gradient updates. During adaptation, the resulting weights ($w_{t,n}$) are integrated into the loss function as described in Equation equation 3. This selective weighting emphasizes critical samples while reducing the influence of less relevant ones, thereby enhancing the model's learning efficiency.

### A.4.2 DETAILED ABLATION STUDIES ON SELECTIVE WEIGHTING.

The average success rate per benchmark is illustrated in Table 2. The detailed results on each task are shown in Table 7, Table 8, and Table 9. Additionally, Figure 10 presents the sensitivity analysis of the hyperparameters—lower threshold ($\frac{1}{8}$), higher threshold ($\frac{1}{3}$), and padding step (15 steps)—

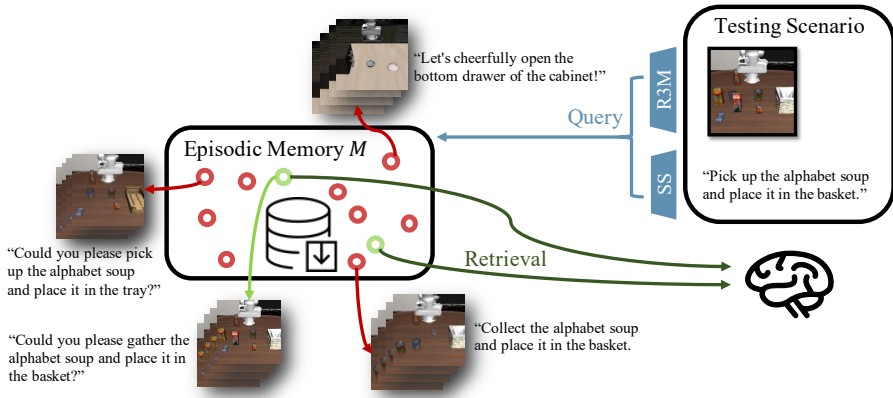

Figure 8: Data Retrieval. Episodic Memory $\mathcal{M}$ randomly stores a few demonstrations collected during lifelong learning. To retrieve a small number of demonstrations most similar to the current scenario, we compute a weighted distance (Eq 2) using both image and language embeddings. In $\mathcal{M}$, red and green circles denote relevant and irrelevant demonstrations, respectively, which include language descriptions, visual observations, and joint and gripper states. The retrieved demonstrations are then used for Weighted Local Adaptation.

---

**Algorithm 1** Task-unaware Retrieval-based Weighted Local Adaptation

---

***Lifelong Learning*:**

    Initialize model parameter $\theta$, episodic memory $\mathcal{M} = \{\}$, and tasks $\{\mathcal{T}_i\}, i = 1, 2, \ldots, T$

    **for** $K \in \{1, 2, \ldots, T\}$ **do**

        Train $\theta$ on $\mathcal{D}_K \cup \mathcal{M}$ using Eq 1

        Randomly store $16\%$ demonstrations from $\mathcal{D}_K$ into $\mathcal{M}$

    **end for**

During deployment, robot encounters a testing scenario $\mathcal{S}_{deploy} \sim p(\mathcal{T}_i), 1 \le i \le T$:

***Reviewing*:**

    Rollout 10 episodes on $\mathcal{S}_{deploy}$ to assess robot's performance with $\theta$;

    Retrieve $\tilde{N}$ demonstrations from $\mathcal{M}$ based on $\mathcal{D}_R$ using Eq 2 (4.1);

    Compute $w_{t,n}$ based on selective weighting (4.2.1);

    $\theta' \leftarrow$ Locally adapt $\theta$ using Eq 3 as skill restoration within limited epochs (4.2.2);

***Final Testing*:**

    Test $\theta'$ in $\mathcal{S}_{deploy}$.

---

used to identify Separation Segments during selective weighting. The experiments are conducted on three random seeds as well. The results demonstrate that our proposed method's performance is robust to variations in these hyperparameters.

## A.5 DETAILED TESTING RESULTS

We selected 20 typical scenarios among `libero_90`. The list of those scenarios can be found in Table 10. Additionally, the testing results of our method and baselines including **ER-WLA**, **ER**, **Packnet**, are listed in Table 11

## A.6 DISCUSSION ON POTENTIAL FORGETTING DURING LOCAL ADAPTATION

Our method addresses this issue through a robust deployment strategy. After sequential learning, we preserve the final model as a stable foundation. For each testing scenario, we fine-tune a copy of this model using our weighted local adaptation mechanism. Crucially, we always return to the preserved final model for subsequent scenarios, ensuring that each adaptation starts from the same

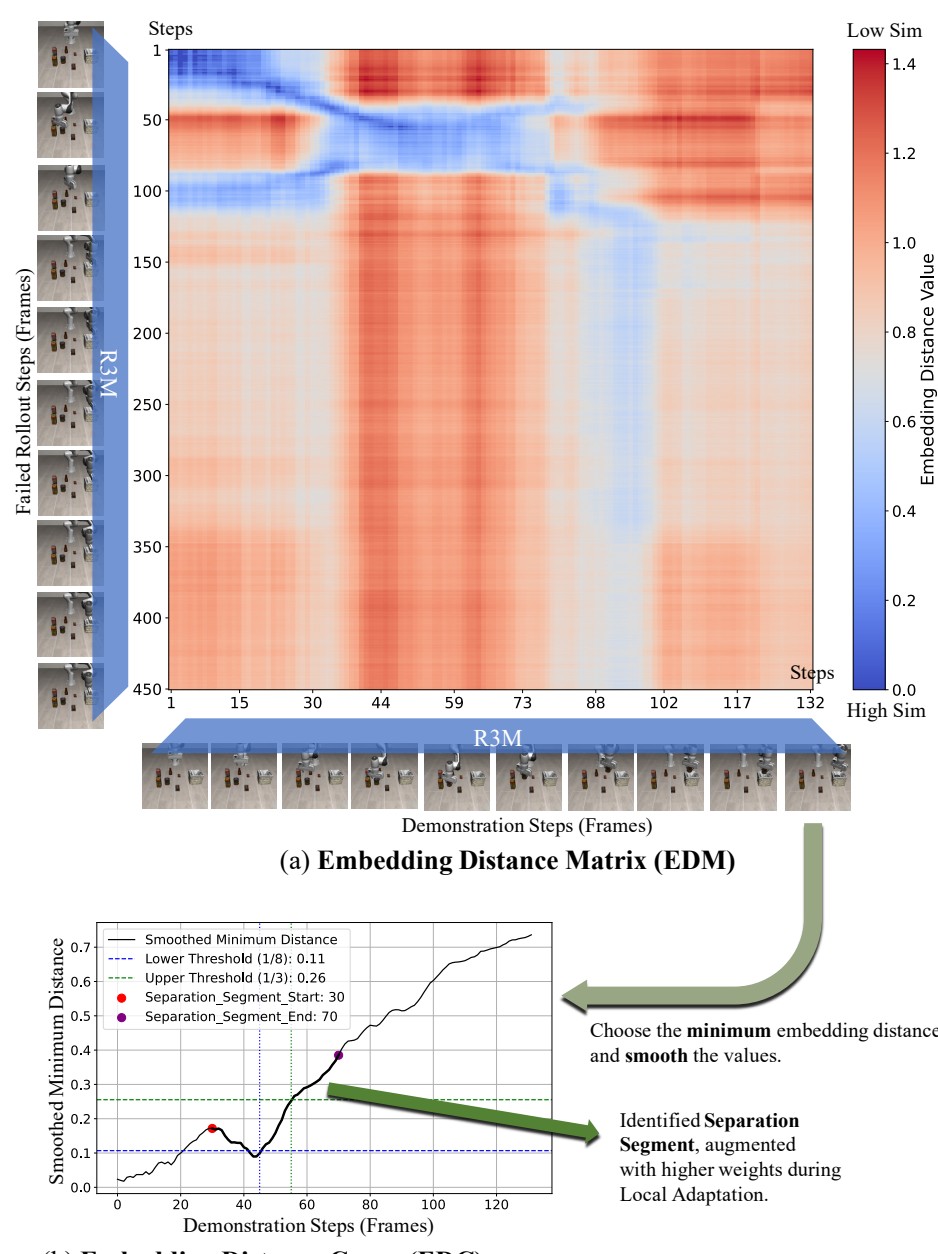

(a) **Embedding Distance Matrix (EDM)**

Choose the **minimum** embedding distance and **smooth** the values.

Identified **Separation Segment**, augmented with higher weights during Local Adaptation.

(b) **Embedding Distance Curve (EDC)**

Figure 9: Illustration of the selective weighting heuristic using (a) **Embedding Distance Matrix (EDM)** and (b) **Embedding Distance Curve (EDC)**. In the demonstration, the robot successfully picks up a jar and places it into a basket. In the failed rollout, the robot fails during the picking stage, resulting in the absence of subsequent steps. The steps surrounding the picking procedure are identified as the **Separation Segment** and are assigned higher weights during adaptation to address the model's shortcomings. Specifically, the Separation Segment is determined by the smoothed minimum $L_2$ distances from EDC—obtained from EDM, where each of its entry indicates the embedding distance between a demonstration and failed rollout frame, as shown in this figure.

well-trained baseline and previous adaptations do not influence future ones. This approach keeps local adaptations isolated and prevents the accumulation of forgetting effects.

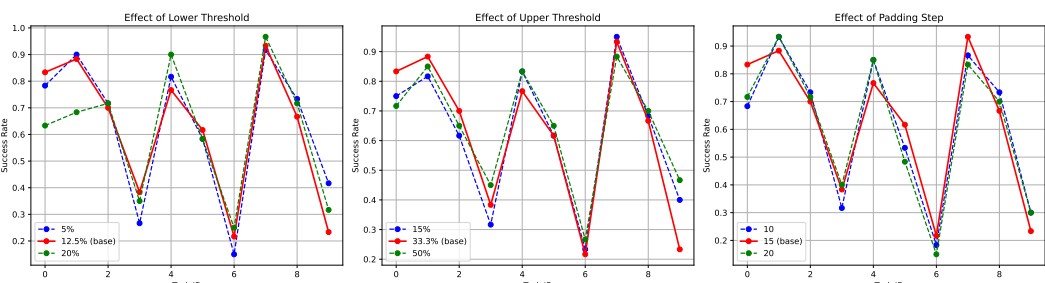

Figure 10: Hyperparameter Sensitivity Check.

Table 7: Ablation Study Results on `libero_object`: Average Success Rates and Standard Deviation for Each Task Across Epochs.

| Method Task Epoch | ULA | | | WLA | | |
|---|---|---|---|---|---|---|
| | Epoch 15 | Epoch 20 | Epoch 25 | Epoch 15 | Epoch 20 | Epoch 25 |
| 0 | $0.68 \pm 0.04$ | $0.37 \pm 0.12$ | $0.62 \pm 0.04$ | $0.57 \pm 0.14$ | $0.67 \pm 0.15$ | $0.57 \pm 0.04$ |
| 1 | $0.20 \pm 0.08$ | $0.40 \pm 0.13$ | $0.35 \pm 0.22$ | $0.35 \pm 0.15$ | $0.45 \pm 0.06$ | $0.13 \pm 0.06$ |
| 2 | $0.77 \pm 0.14$ | $0.85 \pm 0.06$ | $0.78 \pm 0.15$ | $0.90 \pm 0.08$ | $0.78 \pm 0.14$ | $0.82 \pm 0.11$ |
| 3 | $0.68 \pm 0.15$ | $0.78 \pm 0.06$ | $0.70 \pm 0.03$ | $0.70 \pm 0.09$ | $0.60 \pm 0.10$ | $0.75 \pm 0.08$ |
| 4 | $0.75 \pm 0.08$ | $0.87 \pm 0.02$ | $0.78 \pm 0.07$ | $0.70 \pm 0.06$ | $0.78 \pm 0.07$ | $0.88 \pm 0.03$ |
| 5 | $0.47 \pm 0.19$ | $0.65 \pm 0.05$ | $0.53 \pm 0.04$ | $0.37 \pm 0.09$ | $0.42 \pm 0.07$ | $0.60 \pm 0.13$ |
| 6 | $0.52 \pm 0.06$ | $0.53 \pm 0.09$ | $0.38 \pm 0.16$ | $0.65 \pm 0.12$ | $0.52 \pm 0.12$ | $0.55 \pm 0.08$ |
| 7 | $0.47 \pm 0.19$ | $0.58 \pm 0.14$ | $0.57 \pm 0.09$ | $0.58 \pm 0.04$ | $0.73 \pm 0.04$ | $0.60 \pm 0.18$ |
| 8 | $0.55 \pm 0.10$ | $0.58 \pm 0.17$ | $0.50 \pm 0.13$ | $0.58 \pm 0.06$ | $0.70 \pm 0.09$ | $0.72 \pm 0.09$ |
| 9 | $0.70 \pm 0.18$ | $0.45 \pm 0.10$ | $0.58 \pm 0.03$ | $0.40 \pm 0.15$ | $0.58 \pm 0.02$ | $0.53 \pm 0.09$ |

Table 8: Ablation Study Results on libero_goal: Average Success Rates and Standard Deviation for Each Task Across Epochs.

| Method Task Epoch | ULA | | | WLA | | |
|---|---|---|---|---|---|---|
| | Epoch 15 | Epoch 20 | Epoch 25 | Epoch 15 | Epoch 20 | Epoch 25 |
| 0 | $0.62 \pm 0.09$ | $0.75 \pm 0.05$ | $0.68 \pm 0.10$ | $0.72 \pm 0.04$ | $0.83 \pm 0.09$ | $0.65 \pm 0.03$ |
| 1 | $0.88 \pm 0.03$ | $0.92 \pm 0.03$ | $0.88 \pm 0.02$ | $0.87 \pm 0.06$ | $0.88 \pm 0.04$ | $0.92 \pm 0.04$ |
| 2 | $0.65 \pm 0.13$ | $0.72 \pm 0.12$ | $0.80 \pm 0.03$ | $0.68 \pm 0.08$ | $0.70 \pm 0.08$ | $0.83 \pm 0.06$ |
| 3 | $0.38 \pm 0.07$ | $0.25 \pm 0.03$ | $0.32 \pm 0.09$ | $0.32 \pm 0.12$ | $0.38 \pm 0.16$ | $0.32 \pm 0.06$ |
| 4 | $0.88 \pm 0.04$ | $0.80 \pm 0.05$ | $0.82 \pm 0.03$ | $0.87 \pm 0.04$ | $0.77 \pm 0.14$ | $0.92 \pm 0.04$ |
| 5 | $0.60 \pm 0.10$ | $0.53 \pm 0.13$ | $0.63 \pm 0.20$ | $0.58 \pm 0.07$ | $0.62 \pm 0.12$ | $0.77 \pm 0.03$ |
| 6 | $0.15 \pm 0.03$ | $0.15 \pm 0.09$ | $0.22 \pm 0.07$ | $0.13 \pm 0.04$ | $0.22 \pm 0.06$ | $0.20 \pm 0.03$ |
| 7 | $0.93 \pm 0.04$ | $0.95 \pm 0.05$ | $1.00 \pm 0.00$ | $0.97 \pm 0.02$ | $0.93 \pm 0.02$ | $0.93 \pm 0.04$ |
| 8 | $0.78 \pm 0.07$ | $0.80 \pm 0.03$ | $0.77 \pm 0.04$ | $0.72 \pm 0.06$ | $0.67 \pm 0.11$ | $0.90 \pm 0.05$ |
| 9 | $0.25 \pm 0.03$ | $0.33 \pm 0.08$ | $0.50 \pm 0.10$ | $0.43 \pm 0.17$ | $0.23 \pm 0.06$ | $0.32 \pm 0.09$ |

Table 9: Ablation Study Results on libero_spatial: Average Success Rates and Standard Deviation for Each Task Across Epochs.

| Method | ULA | | | WLA | | |
|---|---|---|---|---|---|---|
| Task Epoch | Epoch 15 | Epoch 20 | Epoch 25 | Epoch 15 | Epoch 20 | Epoch 25 |
| 0 | $0.35 \pm 0.18$ | $0.33 \pm 0.07$ | $0.47 \pm 0.07$ | $0.45 \pm 0.10$ | $0.45 \pm 0.10$ | $0.42 \pm 0.13$ |
| 1 | $0.48 \pm 0.09$ | $0.43 \pm 0.04$ | $0.48 \pm 0.10$ | $0.30 \pm 0.05$ | $0.58 \pm 0.19$ | $0.40 \pm 0.13$ |
| 2 | $0.32 \pm 0.11$ | $0.35 \pm 0.13$ | $0.28 \pm 0.12$ | $0.40 \pm 0.19$ | $0.50 \pm 0.13$ | $0.45 \pm 0.13$ |
| 3 | $0.48 \pm 0.03$ | $0.47 \pm 0.09$ | $0.60 \pm 0.05$ | $0.48 \pm 0.07$ | $0.47 \pm 0.11$ | $0.50 \pm 0.00$ |
| 4 | $0.17 \pm 0.04$ | $0.30 \pm 0.03$ | $0.13 \pm 0.07$ | $0.22 \pm 0.07$ | $0.18 \pm 0.07$ | $0.23 \pm 0.02$ |
| 5 | $0.12 \pm 0.09$ | $0.20 \pm 0.08$ | $0.28 \pm 0.09$ | $0.25 \pm 0.10$ | $0.22 \pm 0.09$ | $0.27 \pm 0.02$ |
| 6 | $0.60 \pm 0.13$ | $0.58 \pm 0.02$ | $0.47 \pm 0.10$ | $0.57 \pm 0.07$ | $0.58 \pm 0.07$ | $0.67 \pm 0.03$ |
| 7 | $0.52 \pm 0.06$ | $0.42 \pm 0.02$ | $0.38 \pm 0.07$ | $0.38 \pm 0.06$ | $0.38 \pm 0.04$ | $0.38 \pm 0.06$ |
| 8 | $0.30 \pm 0.05$ | $0.42 \pm 0.08$ | $0.30 \pm 0.00$ | $0.40 \pm 0.10$ | $0.28 \pm 0.03$ | $0.30 \pm 0.03$ |
| 9 | $0.20 \pm 0.10$ | $0.32 \pm 0.09$ | $0.42 \pm 0.03$ | $0.17 \pm 0.07$ | $0.33 \pm 0.06$ | $0.17 \pm 0.04$ |

Table 10: Selected Tasks for *libero_different_scenes* benchmark from *libero_90*

| Task ID | Initial Descriptions | Scenes |
|---|---|---|
| 1 | Close the top drawer of the cabinet | Kitchen scene10 |
| 2 | Open the bottom drawer of the cabinet | Kitchen scene1 |
| 3 | Open the top drawer of the cabinet | Kitchen scene2 |
| 4 | Put the frying pan on the stove | Kitchen scene3 |
| 5 | Close the bottom drawer of the cabinet | Kitchen scene4 |
| 6 | Close the top drawer of the cabinet | Kitchen scene5 |
| 7 | Close the microwave | Kitchen scene6 |
| 8 | Open the microwave | Kitchen scene7 |
| 9 | Put the right moka pot on the stove | Kitchen scene8 |
| 10 | Put the frying pan on the cabinet shelf | Kitchen scene9 |
| 11 | Pick up the alphabet soup and put it in the basket | Living Room scene1 |
| 12 | Pick up the alphabet soup and put it in the basket | Living Room scene2 |
| 13 | Pick up the alphabet soup and put it in the tray | Living Room scene3 |
| 14 | Pick up the black bowl on the left and put it in the tray | Living Room scene4 |
| 15 | Put the red mug on the left plate | Living Room scene5 |
| 16 | Put the chocolate pudding to the left of the plate | Living Room scene6 |
| 17 | Pick up the book and place it in the front compartment of the caddy | Study scene1 |
| 18 | Pick up the book and place it in the back compartment of the caddy | Study scene2 |
| 19 | Pick up the book and place it in the front compartment of the caddy | Study scene3 |
| 20 | Pick up the book in the middle and place it on the cabinet shelf | Study scene4 |

Table 11: Detailed Comparisons on *libero_different_scenes* Benchmark. It illustrates that after reaching the capacity of PackNet, it could no longer deal with new tasks anymore. Besides, the task-specific results for Experience Replay (ER) and proposed retrieval-based weighted local adaptation (ER-WLA) also show 1) consistently low variance within individual tasks, 2) 16 (out of 20) tasks' performance has been raised with ER-WLA, both demonstrating our method's stability and effectiveness.

| Task | ER-WLA | ER | Packnet |
|------|--------|-----|---------|
| 0 | $0.85 \pm 0.08$ | $0.50 \pm 0.03$ | $1.00 \pm 0.00$ |
| 1 | $0.13 \pm 0.08$ | $0.27 \pm 0.06$ | $0.83 \pm 0.09$ |
| 2 | $0.73 \pm 0.09$ | $0.72 \pm 0.10$ | $0.92 \pm 0.02$ |
| 3 | $0.40 \pm 0.03$ | $0.13 \pm 0.02$ | $0.17 \pm 0.03$ |
| 4 | $0.93 \pm 0.04$ | $0.72 \pm 0.10$ | $1.00 \pm 0.00$ |
| 5 | $1.00 \pm 0.00$ | $0.57 \pm 0.16$ | $1.00 \pm 0.00$ |
| 6 | $0.52 \pm 0.04$ | $0.52 \pm 0.03$ | $0.78 \pm 0.04$ |
| 7 | $0.82 \pm 0.07$ | $0.63 \pm 0.09$ | $0.88 \pm 0.02$ |
| 8 | $0.32 \pm 0.07$ | $0.23 \pm 0.06$ | $0.00 \pm 0.00$ |
| 9 | $0.48 \pm 0.15$ | $0.38 \pm 0.12$ | $0.00 \pm 0.00$ |
| 10 | $0.23 \pm 0.06$ | $0.03 \pm 0.02$ | $0.00 \pm 0.00$ |
| 11 | $0.20 \pm 0.03$ | $0.10 \pm 0.06$ | $0.00 \pm 0.00$ |
| 12 | $0.23 \pm 0.09$ | $0.13 \pm 0.02$ | $0.00 \pm 0.00$ |
| 13 | $0.67 \pm 0.09$ | $0.83 \pm 0.04$ | $0.00 \pm 0.00$ |
| 14 | $0.15 \pm 0.03$ | $0.13 \pm 0.04$ | $0.00 \pm 0.00$ |
| 15 | $0.68 \pm 0.09$ | $0.30 \pm 0.08$ | $0.00 \pm 0.00$ |
| 16 | $0.03 \pm 0.03$ | $0.00 \pm 0.00$ | $0.00 \pm 0.00$ |
| 17 | $0.28 \pm 0.08$ | $0.02 \pm 0.02$ | $0.00 \pm 0.00$ |
| 18 | $0.10 \pm 0.08$ | $0.02 \pm 0.02$ | $0.00 \pm 0.00$ |
| 19 | $0.27 \pm 0.16$ | $0.58 \pm 0.07$ | $0.00 \pm 0.00$ |

