# OpenReview forum: "Task-Unaware Lifelong Robot Learning with Retrieval-based Weighted Local Adaptation"
_ICLR.cc/2025/Conference — Submitted to ICLR 2025_

### Official Review · Reviewer_qCFh · 2024-11-01

**Soundness:** 4
**Presentation:** 4
**Contribution:** 3
**Rating:** 8
**Confidence:** 5

**Summary:**

The paper investigates the problem of lifelong robot learning where catastrophic forgetting is a key issue. The paper proposes to do retrieval-based adaptation in lifelong learning process, finetuning the global policy model with memories of previous similar tasks. Despite the simplicity of the idea, the performance exceeds multiple baselines in the experiments. The paper is well-written, and the results support the claims well. I do have a few detailed questions hoping the authors could clarify during the discussion period.

**Strengths:**

1. The paper is well written and easy to follow, with the figures illustrative about core ideas.
2. The idea proposed is simple yet effective, showing great potential.
3. The results and benchmark comparisons are very well structured, clearly showing the support for certain questions outlined in Section 5. The ablation studies are also well structured with both quantitative and qualitative evidence.
4. The introduction provides strong motivation for the lifelong robot learning problem.

**Weaknesses:**

1. Although the related work section is comprehensive, it lacks a clear description of how the current work distinguishes itself from the related work. I suggest authors add this information and the relationship between the current work with prior work in each subsection of the related work.

1.1. Another work that looks into lifelong robot learning from demonstration is [1]. Could the authors possibly compare with it either conceptually or empirically?

[1] Chen, L., Jayanthi, S., Paleja, R. R., Martin, D., Zakharov, V., & Gombolay, M. (2023, March). Fast lifelong adaptive inverse reinforcement learning from demonstrations. In Conference on Robot Learning (pp. 2083-2094). PMLR.

2. The description about “blurred task boundaries” a little misleading. One would have thought blurred task boundaries mean there are no clear temporal boundaries between different task executions. However, the paper seems to take language-described robot tasks as blurred tasks as opposed to task labels, and there are clearly separated demonstrations among different tasks. I hope the authors could make the definition of “blurred task boundaries” clearer.

3. I also find some parts of the methods are not precise. For example, it is unclear from Section 4.1 whether the retrieval is on the timestep-level or demonstration-level. It became clearer with later discussion that it should be demonstration-level, but adding a clearer pseudocode of the entire pipeline would be helpful.

4. The abstract mentioned storing past data to prevent forgetting has privacy concerns. However, according to my understanding, the proposed method also requires storing the demonstrations in memory M for retrieval, thus having a similar problem. Can the authors discuss how this is handled in the proposed method?

**Questions:**

Besides the points mentioned in weaknesses,
1. How are the demonstrations generated? Did humans provide demonstrations or was the demonstrations generated by a well-trained RL policy?
2. How are the hyperparameters tuned, especially the language and image distance weights $\alpha_v$ and $\alpha_l$?

---

> ### Author Response · Authors · 2024-11-24
> **Response to Reviewer qCFh**
>
> Thank you so much for your insightful feedback!
>
> >  Although the related work section is comprehensive, it lacks a clear description of how the current work distinguishes itself from the related work. I suggest authors add this information and the relationship between the current work with prior work in each subsection of the related work.
>
> We agree our initial version of related works wasn’t comprehensive and thank you for your suggestions! We have revised the related work section to clearly distinguish our approach from existing literature, and included the paper by Chen et al. (2023) in **section 2.1** at lines 110\~113. Below is a concise conceptual comparison with the paper (Chen et al. (2023)):
>
>
> Both our work and FLAIR address lifelong robot adaptation without relying on task or strategy labels. However, there are key distinctions:
>
> **Objective Focus**:
> - FLAIR primarily aims for efficient lifelong learning from demonstrations, especially when sequentially encountered demonstrations are heterogeneous or have user-specific preferences. And it focuses on forward transferring the knowledge. Due to its reliance on non-stationary reward functions from AIRL, FLAIR may experience some degree of forgetting.
> - Our Approach proactively concentrates on mitigating catastrophic forgetting during the robot's lifelong learning. We focus on preserving and restoring previously learned skills and knowledge to maintain model stability over time.
>
> **Adaptation Mechanism**:
> - FLAIR improves efficiency and scalability by representing new demonstrations by mixing previously trained strategies, rather than creating new policies for each incoming demonstration. It employs Multi-Strategy Reward Distillation (MSRD) and Between-Class Discrimination (BCD) to update strategies on the fly, distilling common knowledge and distinguishing different strategies.
> - Our Method involves the robot rolling out several episodes to assess any forgetting in the model for previously learned but forgotten scenarios. It then retrieves related demonstrations from episodic memory and applies weighted local adaptation to rapidly restore the learned skills and knowledge.
>
>
>
> >  The description about “blurred task boundaries” a little misleading. One would have thought blurred task boundaries mean there are no clear temporal boundaries between different task executions. However, the paper seems to take language-described robot tasks as blurred tasks as opposed to task labels, and there are clearly separated demonstrations among different tasks. I hope the authors could make the definition of “blurred task boundaries” clearer.
>
> Thank you for pointing this out! We agree that our description of "blurred task boundaries" in the first submission was unclear. To address this, we have revised the definition and explanation in **Preliminary Section (Section 3)** at lines 192\~ 207. We've also refined related explanations in the introduction.
>
>
>
>
>
> >  I also find some parts of the methods are not precise. For example, it is unclear from Section 4.1 whether the retrieval is on the timestep-level or demonstration-level. It became clearer with later discussion that it should be demonstration-level, but adding a clearer pseudocode of the entire pipeline would be helpful.
>
> Thank you again for your suggestion about this! We have updated the paper to make the retrieval procedure and the whole pipeline clearer. Specifically, we mentioned our pseudocode of entire pipeline at **Section 4** at lines 229\~230 and visualize the retrieval process at  **Section 4.1** at lines 259\~260. (The pseudocode and retrieval procedure visualization are **Algorithm 1** at lines 993\~1008 and **Figure 8** at lines 972\~991 in Appendix)
>
>
>
> > The abstract mentioned storing past data to prevent forgetting has privacy concerns. However, according to my understanding, the proposed method also requires storing the demonstrations in memory M for retrieval, thus having a similar problem. Can the authors discuss how this is handled in the proposed method?
>
>
> Thank you for your insightful comment. We acknowledge that storing past data can raise privacy concerns. Our proposed method mitigates this issue by randomly storing only a small number of demonstrations in the episodic memory (as shown in **Algorithm 1** at lines 993\~1008). This limited storage reduces the potential for privacy issues compared to methods that retain all past data.
>
> Although we did not discuss privacy considerations in detail in the paper, our framework is designed to be extensible and can incorporate privacy-preserving mechanisms. For instance, it can be enhanced with a privacy filtering module that selectively stores only privacy-compliant demonstrations.

---

> ### Author Response · Authors · 2024-11-24
> **Response to Reviewer qCFh**
>
> > How are the demonstrations generated? Did humans provide demonstrations or was the demonstrations generated by a well-trained RL policy?
>
> Following the design from LIBERO benchmarks, we use demonstrations generated by humans.
>
> Liu, Bo, et al. "Libero: Benchmarking knowledge transfer for lifelong robot learning." Advances in Neural Information Processing Systems 36 (2024).
>
>
> > How are the hyperparameters tuned, especially the language and image distance weights $\alpha_1$ and $\alpha_2$?
>
> We have conducted additional experiments (see **Figure 10** at lines 1082\~1093) that demonstrate the robustness of our method to the threshold hyperparameters used in selective weighting.
>
> Regarding the distance weights $\alpha_1$ and $\alpha_2$, the two hyperparameters are designed to improve the retrieval of relevant demonstrations for skill restoration during local adaptation.
>
> In the LIBERO benchmark tasks, in which we further paraphrased language descriptions, we observed issues with task overlapping due to blurred task boundaries—both visually and linguistically (as illustrated in **Figures 7 and 8** at lines 937\~951, 972\~991). To address this, we adjust the weights $\alpha_1$ and $\alpha_2$ to tune the reliability of each modality. If one aspect (visual or language) leads to confusion, we reduce its impact on the retrieval process by lowering its corresponding $\alpha$ value.
>
> **Noteworthy**, our experimental results using BERT as the language model (shown in **Figure 4 (c)** at lines 514\~528) indicate that even with a relatively low retrieval success rate (above 10%), our weighted local adaptation still enhances performance. This demonstrates that our approach is fault-tolerant and robust, not heavily dependent on precise data retrieval.
>
>
> Please let us know if the replies answered your questions! Thank you!

---

> > ### Comment · Reviewer_qCFh · 2024-11-24
> >
> > Thank you to the authors for very detailed replies to all my concerns. I believe all my questions and concerns have been addressed with the answers and edits of the paper. Therefore, I have raised my score to be 8: accept, good paper.

---

> > > ### Author Response · Authors · 2024-11-24
> > > **Thank you for increasing the score**
> > >
> > > We sincerely thank you for the response and increasing the score!

---

### Official Review · Reviewer_mMan · 2024-11-03

**Soundness:** 2
**Presentation:** 3
**Contribution:** 3
**Rating:** 5
**Confidence:** 2

**Summary:**

This work develops a novel approach for robotic control in a lifelong learning, multitask context without clear boundaries between tasks, suitable for real-world use. It uses episodic memory for experience replay and testing, allowing rapid adaptation to previously encountered tasks without explicit task identifiers. It is experimentally validated on the LIBERO benchmark and shows a significant improvement over baseline methods.

**Strengths:**

This paper focuses discusses a very important topic, that is robot learning in scenarios where there is no sharp boundaries between tasks, which is important for robots that operate in complex and noisy real world environments. The method is technically sound, and the empirical evaluation shows very good results.

**Weaknesses:**

My main concern is the magnitude of the uncertainties in Table 1. One of the main results is a difference between 45.17 ± 31.86 and 34.08 ± 28.55, which does not seem statistically significant - similar problems appear in the other results as well.

Potentially related to that, lines 402-403 state that the algorithm is evaluated on three seeds: 1, 21, 42. This is a suspiciously arbitrary selection - why not 1, 2, 3? This makes me a bit concerned that the seeds were cherry-picked to select for the most convenient results.

**Questions:**

What is the practical computational cost of this method? As in, if I wanted to train a model using this approach, assuming I have all the code and hyperparameters, what kind of hardware and time would I need to go from nothing to replicating the paper's results?

Is it viable to run additional experiments to tighten the uncertainties reported in Tables 1 and 2? As I understand, the standard deviation is based on the three seeds, and with a standard deviation this large, a larger sample and a proper statistical analysis would be very valuable.

Can you comment on why the selected seeds are 1, 21 and 42?


If my concern about the uncertainty and potentially cherry-picking the seeds is answered, I will be happy to increase my rating.

---

> ### Author Response · Authors · 2024-11-23
> **Response to Reviewer mMan**
>
> Thank you for your thorough review and important questions about our experimental setup and results!
>
> > What is the practical computational cost of this method? As in, if I wanted to train a model using this approach, assuming I have all the code and hyperparameters, what kind of hardware and time would I need to go from nothing to replicating the paper's results?
>
> While the lifelong learning phase requires approximately 15 hours on an NVIDIA A100 for sequentially training 10 tasks, the “reviewing" phase is highly efficient - with parallel rollout taking fewer than 1 minute and weighted local adaptation requiring around 1 minute.
>
> > My main concern is the magnitude of the uncertainties in Table 1. One of the main results is a difference between 45.17 ± 31.86 and 34.08 ± 28.55, which does not seem statistically significant - similar problems appear in the other results as well.
>
> Thank you for your question and we appreciate this opportunity to clarify the results with high variances.
>
> The high standard deviations (e.g., 45.17 ± 31.86 vs 34.08 ± 28.55) stem from task diversity rather than method instability. Different tasks naturally vary in complexity - (as shown in the corresponding task-specific results from **Table 11 in Appendix** at lines 1200\~1230) some achieve success rates around 0.9, while more challenging ones may reach 0.2 - leading to a high variance for the Average Success Rate (ASR) over all the tasks. Table 11's results (over the three random seeds) also show
> - consistently low variance within individual tasks
> - 16 (out of 20) tasks’ performance has been raised in the *libero_different_scenes* benchmark with the proposed weighted local adaptation (ER-WLA)
>
> demonstrating our method's stability and effectiveness. We also added the explanation to our updated paper in **section 5.1.3 from lines 378 to 381**.
>
> > Can you comment on why the selected seeds are 1, 21 and 42?
>
> We simply chose the seeds based on a very relevant continual robot learning work: "TAIL: Task-specific Adapters for Imitation Learning with Large Pretrained Models" by Zuxin Liu et al. (ICLR 2024), which used seed values (0, 21, 42) in their experimental setting. **This was not an instance of cherry-picking but just a decision aimed at comparability.**
>
>
> Thank you so much again and look forward to your reply!

---

> ### Author Response · Authors · 2024-12-01
> **Reply to Reviewer mMan**
>
> Dear reviewer mMan,
>
> Thank you again for your constructive comments! As we are near the end of the discussion period, we hope to know whether our responses have addressed your concerns. We look forward to your further feedback and suggestions.
>
> Best regards,
>
> Authors of 12126

---

> ### Author Response · Authors · 2024-12-03
> **Reminder for the response**
>
> > My main concern is the magnitude of the uncertainties in Table 1. One of the main results is a difference between 45.17 ± 31.86 and 34.08 ± 28.55, which does not seem statistically significant - similar problems appear in the other results as well.
>
> Based on your suggestion, we conducted additional key experiments using seeds 2 and 3 on all four benchmarks to compare our proposed method ER-WLA with the ER baseline (serving as the backbone of our approach and is representative of SOTA memory-based lifelong learning methods).
>
> The updated results, now averaged over five random seeds (1, 2, 3, 21, 42), are as follows:
>
> | Benchmarks\Methods      |       **ER**      |       **ER-WLA** |
> |:------------------------:|:---------------:|:----------------:|
> | libero_spatial                   |       15.6 ± 12.27        |             **39.3    ±  19.43**    |
> | libero_object | 53.9 ± 21.70 | **57.2  ± 21.36** |
> | libero_goal | 53.1 ± 22.49 | **61.5 ± 28.54** |
> | libero_different_scenes | 35.8 ± 29.44 | **44.95 ± 32.48** |
>
>
> It demonstrates that ER-WLA outperforms the baseline, providing more significant statistical evidence with more seeds. Furthermore, as explained in our previous reply, task diversity leads to high variances, which is also evident in these additional results—increasing the samples did not tighten the uncertainties.
>
>
> Thank you again for your thoughtful feedback! We hope the explanations and additional results addressed your concerns and would greatly appreciate it if you could kindly consider updating your rating.

---

### Official Review · Reviewer_UrfP · 2024-11-04

**Soundness:** 3
**Presentation:** 3
**Contribution:** 4
**Rating:** 5
**Confidence:** 3

**Summary:**

This paper presents a framework for lifelong robot learning that operates without explicit task boundaries or identifiers. The approach combines retrieval-based adaptation with selective weighting to help robots maintain and restore previously learned skills. The method uses episodic memory for both experience replay during training and local adaptation during testing. When encountering a task, the system retrieves relevant past demonstrations based on visual and language similarities, identifies challenging segments through preliminary rollouts, and applies weighted local adaptation. The approach is evaluated on LIBERO benchmark variants.

**Strengths:**

1. Originality:
- Novel approach to handling task boundaries without explicit task IDs
- Interesting combination of retrieval-based adaptation and selective weighting
- Dual use of episodic memory for training and testing phases

2. Quality:
- Comprehensive experimental evaluation across benchmarks
- Detailed ablation studies
- Clear documentation of implementation details

3. Clarity:
- Well-structured presentation
- Clear figures and visualizations
- Detailed appendices

4. Significance:
- Addresses a relevant challenge in robotics
- Shows potential for generalization across memory-based approaches
- Demonstrates improvements over baselines in controlled settings

**Weaknesses:**

1. Conceptual Contradiction:
- Claims to be "task-unaware" but fundamentally relies on task-based retrieval
- Still requires matching current scenarios with previously stored demonstrations
- The evaluation is still conducted on clearly separated benchmark tasks

2. Experiment:
- All results are from simulation with no real-world validation
- No analysis of failure modes or edge cases

3. Technical:
- Limited discussion of the impact of different hyperparameters
- Memory requirements could become problematic with increasing task numbers
- No discussion of potential catastrophic forgetting during local adaptation

**Questions:**

- How is your approach fundamentally different from traditional task-based methods, given that it still relies on retrieving similar tasks?
- What happens when encountering truly novel scenarios that don't match any stored demonstrations?
- How do you justify the various manual thresholds in the selective weighting mechanism? Have you explored their sensitivity?

---

> ### Author Response · Authors · 2024-11-24
> **Response to Reviewer UrfP (part 1)**
>
> Thank you so much for your insightful feedback!
>
> > Conceptual Contradiction: Claims to be "task-unaware" but fundamentally relies on task-based retrieval (How is your approach fundamentally different from traditional task-based methods, given that it still relies on retrieving similar tasks?)
>
>
> Thank you for highlighting this concern. We recognize that some concepts were not clearly defined in our initial submission and have updated the paper accordingly. We provide the following explanations to better clarify the distinctions between our task-unaware approach and traditional task-based methods.
>
> **Our method is fundamentally task-unaware because it operates without the need for explicit task identifiers or implicit task boundaries** (We have updated our paper in **Section 3** at lines 190\~207). Specifically, our approach applies
>
> - experience replay (ER) during training,
> - demonstration-based retrieval during deployment (for weighted local adaptation), purely based on similarity in visual observations and language descriptions
>
> without any task classification or identification. In contrast, many other continual learning algorithms—such as some regularization-based (e.g., EWC computes the Fisher Information Matrix at task boundaries) and some dynamic-architecture-based methods (e.g., PackNet requires task IDs to apply task-specific masks)—are task-based. Our experimental results (**Section 5.2.2 at lines 473\~476**) further show that our weighted local adaptation is robust and fault-tolerant, even with irrelevant demonstrations retrieved.
>
> Determining which task generated a given scenario is also challenging due to overlapping distributions in environmental settings or language descriptions among different tasks. As demonstrated in **Figure 7** (lines 937\~951), even advanced models like BERT and Sentence Similarity cannot reliably distinguish between tasks based on language descriptions. **Figure 8** (lines 972\~992) further shows that our retrieval mechanism is based on demonstration similarity, while it also highlights that substantial environmental and language similarities exist across different tasks.
>
> We hope this clarification addresses your concern, and we appreciate your feedback in helping us improve the clarity of our work.
>
>
>
>
>
>
>
>
>
> > Conceptual Contradiction: The evaluation is still conducted on clearly separated benchmark tasks
>
> Our primary focus is on restoring learned but forgotten knowledge through a fast review procedure using weighted local adaptation. Therefore, we conducted evaluations on learned scenarios, following the standard experimental setup of the LIBERO benchmark to ensure comparability with established practices.
>
> Additionally, prior research in task-free lifelong learning, such as Aljundi et al. (2019) [1], also utilized experiments with separate tasks—for example, evaluating an agent's obstacle avoidance accuracy in different corridors separately.
>
> To address concerns about task separation, we intentionally blurred task boundaries to create overlapping task distributions, making it more challenging to distinguish between tasks during both training and testing phases. Such tasks’ overlaps are illustrated in **Figures 7 and 8** in the appendix (lines 937\~951 and 972\~991), which also shows that our testing scenario is not clearly separated from different benchmark tasks.
>
> [1] Aljundi, Rahaf, Klaas Kelchtermans, and Tinne Tuytelaars. "Task-free continual learning." Proceedings of the IEEE/CVF Conference on Computer Vision and Pattern Recognition. 2019.
>
>
> > Experiment: All results are from simulation with no real-world validation
>
> We agree with the comment that real-world experiments would better validate our method.
>
> We would like to highlight that we utilized LIBERO, a comprehensive robotic manipulation benchmark specifically designed to closely emulate real-world scenarios and widely recognized in lifelong robot learning research. LIBERO encompasses a diverse range of human daily activities and is built upon robosuite, which provides high-fidelity simulations that accurately capture real-world interactions and dynamics.
>
> We plan to conduct field experiments in our future work for a more comprehensive analysis.
>
>
>
>
> > Experiment: No analysis of failure modes or edge cases
>
> As demonstrated in our experimental results using BERT (See **Section 5.2.2 at lines 473\~476**), even when the most relevant demonstrations were not retrieved, our weighted local adaptation still improved performance. This indicates that our method possesses a degree of fault-tolerance and robustness in handling imperfect retrievals.
>
> Besides, as discussed in **Section 6** (lines 532–536), factors such as manipulation action multimodality can make it challenging to accurately identify separation segments in demonstrations. This can lead to incorrect weighting—an edge case where our method may not perform optimally.

---

> ### Author Response · Authors · 2024-11-24
> **Response to Reviewer UrfP (part 2)**
>
> > Technical: Limited discussion of the impact of different hyperparameters (How do you justify the various manual thresholds in the selective weighting mechanism? Have you explored their sensitivity?)
>
> We agree that our first submission lacks an analysis of impact of hyperparameters. Thank you for pointing this out!
>
> We have updated **Section A.4.1** (Appendix, Pages 17–18) and added **Figure 9** (Appendix, Page 20) to provide a clearer explanation and visual illustration of the heuristic behind the selective weighting mechanism. Furthermore, to provide quantitative validation of the threshold hyperparameters, we have expanded our **Appendix A.4.2** at lines 969\~971, 1011\~1013  with **a new sensitivity analysis experiment**. The results (**Figure 10** in Appendix) demonstrate that our proposed method’s performance is robust to variations in these hyperparameters.
>
>
>
>
> > Technical: No discussion of potential catastrophic forgetting during local adaptation
>
> Thank you for bringing up this important concern regarding potential catastrophic forgetting during local adaptation.
>
> Our method addresses this issue through a robust deployment strategy. After sequential learning, we preserve the final model as a stable foundation. For each testing scenario, we fine-tune a copy of this model using our weighted local adaptation mechanism. Crucially, we always return to the preserved final model for subsequent scenarios, ensuring that each adaptation starts from the same well-trained baseline and previous adaptations do not influence future ones. This approach keeps local adaptations isolated and prevents the accumulation of forgetting effects.
>
> We have added this discussion to **Appendix A.6** on pages 19–20 of the updated paper.
>
>
>
>
>
>
> > Question: What happens when encountering truly novel scenarios that don't match any stored demonstrations?
>
> Our research specifically targets the challenge of catastrophic forgetting in lifelong learning, where the primary goal is maintaining performance on previously encountered skills while learning new ones. The testing environments in our framework are intentionally designed to share similarities with previous demonstrations, as this aligns with our core objective of skill retention.
>
> While our method might retrieve less relevant demonstrations when faced with completely novel scenarios, this could be a future work of our exploration. The fundamental purpose of our work is to ensure robots can reliably maintain and execute previously learned skills, rather than generalizing to entirely new tasks.

---

> ### Author Response · Authors · 2024-11-27
> **Response to Reviewer UrfP (part 3)**
>
> > Technical: Memory requirements could become problematic with increasing task numbers
>
> We have to admit that our method requires storing a small number of the demonstrations, which is problematic when encountering a large number of tasks.
>
> However, as we are only storing a small subset of data in the episodic memory, our method has reduced memory requirements and provides an inherent benefit of reducing privacy concerns through minimal data retention.
>
> Furthermore, since image embeddings—serving dual purposes (input to the manipulation policy and retrieval for local adaptation)—are generated by a **pretrained R3M model (no end-to-end training), our approach is naturally extendable**: this allows for significant storage reduction in future implementations, by simply storing smaller embeddings instead of raw images in the episodic memory.
>
> Thank you again for highlighting this important consideration!
>
>
>
> Let us know if you have any concerns and look forward to your reply!

---

> ### Author Response · Authors · 2024-12-01
> **Reply to Reviewer UrfP**
>
> Dear reviewer UrfP,
>
> Thank you again for your constructive comments! As we are near the end of the discussion period, we hope to know whether our responses have addressed your concerns. We look forward to your further feedback and suggestions.
>
> Best regards,
>
> Authors of 12126

---

> ### Author Response · Authors · 2024-12-03
> **Reminder for the response**
>
> Dear Reviewer UrfP,
>
> As the discussion period is nearing its end, we would like to confirm whether our replies and the updates to the submission have addressed your concerns. **If you have any further comments, we are happy to reply to them before the author response deadline on December 3 (AOE)**.
>
> Thank you again for your insightful suggestions!
>
> Best regards,
> Authors of Paper 12126

---

### Official Review · Reviewer_z13d · 2024-11-05

**Soundness:** 1
**Presentation:** 1
**Contribution:** 1
**Rating:** 5
**Confidence:** 4

**Summary:**

This paper presents an imitation learning approach that enables a single network to learn from demonstrations of multiple tasks. It uses a vision language model and a local weighting mechanism to adapt the network towards the target task. The agent performs a few rollout episodes to assess policy performance, using this feedback for automatic selective weighting by comparing the rollouts with retrieved demonstrations without human intervention. The policy is subsequently adjusted to better align with the desired demonstrations.

**Strengths:**

1. The key idea has some merit, with some analogues to meta-learning approaches
2. The related work is an interesting compilation of papers
3. Figure 3 is well-illustrated, aiding in the exposition of the paper's ideas.
4. PCA Visualization based on SS Model is quite interesting (Figure 4b)

**Weaknesses:**

1. I have concerns regarding the overall presentation of the paper. While the idea holds merit as a novel imitation learning approach, the contributions are significantly overstated. In fact, I would argue that the paper does not fully address a lifelong learning setting. The authors should consider moderating their claims and clearly articulating the scope of their contribution.
2. Many keywords are mentioned but not defined. For instance, what is meant by lifelong learning in this context? Evaluating a lifelong learning agent over only 20 episodes, as done by the authors, seems insufficient and misaligned with the intended concept.
3. The discussion on unspecified task boundaries in real-world scenarios is also unclear and mischaracterized. Although the paper replaces a one-hot task encoding with an embedding from a vision-language model, this essentially still provides a form of task specification, albeit in a more implicit or 'fuzzy' manner.
4. The related work section omits many papers that are highly relevant to the context. While the authors do include a few key references, surprisingly, there isn’t a single paper cited from more than five years ago. I'd recommend the authors to look at the following: [1-6]
5. There is no comparison of the concepts to meta-learning approaches such as [2]. I would argue that their approach resembles having a global network that is fine-tuned to the desired task during testing.
6. The description of the experiments and their results is vague and difficult to follow. While I am confident that the authors are skilled and fully understand their work, I encourage them to consider readers like myself, who may be less familiar with it, by providing a more detailed explanation of their setup. For example, what specific tasks are being solved? Are the demonstrations focused on particular tasks? Does each task in the testing phase have a corresponding relevant demonstration?
7. While the authors claim lifelong learning is important—a sentiment I share—it is unclear why their chosen evaluation setup suits this purpose. They could provide stronger justification for their design choices.


**References**
1. Finn, C., Abbeel, P., & Levine, S. (2017, July). Model-agnostic meta-learning for fast adaptation of deep networks. In International conference on machine learning (pp. 1126-1135). PMLR.
2. Finn, C., Yu, T., Zhang, T., Abbeel, P., & Levine, S. (2017, October). One-shot visual imitation learning via meta-learning. In Conference on robot learning (pp. 357-368). PMLR.
3. Thrun, S. (1995, January). A lifelong learning perspective for mobile robot control. In Intelligent robots and systems (pp. 201-214). Elsevier Science BV.
4. Aljundi, R., Babiloni, F., Elhoseiny, M., Rohrbach, M., & Tuytelaars, T. (2018). Memory aware synapses: Learning what (not) to forget. In Proceedings of the European conference on computer vision (ECCV) (pp. 139-154).
5. Aljundi, R., Kelchtermans, K., & Tuytelaars, T. (2019). Task-free continual learning. In Proceedings of the IEEE/CVF conference on computer vision and pattern recognition (pp. 11254-11263).
6. Grollman, D. H., & Jenkins, O. C. (2007, April). Dogged learning for robots. In Proceedings 2007 IEEE International Conference on Robotics and Automation (pp. 2483-2488). Ieee.

**Questions:**

1. Please refer to the weaknesses section for responses to earlier questions.
2. Would you consider this approach a meta-learning method that aligns with the standard episodic reinforcement learning framework?
3. Isn’t it true that the agent is aware of the task it’s solving, meaning there’s no true task boundary in the proposed evaluation?
4. How would this approach perform if only partial task demonstrations were available? Could it potentially combine these fragments to form a complete solution?

---

> ### Author Response · Authors · 2024-11-25
> **Response to Reviewer z13d (part 1)**
>
> We thank the reviewers' thorough review and important questions!
>
>
> > I have concerns regarding the overall presentation of the paper. While the idea holds merit as a novel imitation learning approach, the contributions are significantly overstated. In fact, I would argue that the paper does not fully address a lifelong learning setting. The authors should consider moderating their claims and clearly articulating the scope of their contribution. (Many keywords are mentioned but not defined. For instance, what is meant by lifelong learning in this context? )
>
>
> We are not quite sure what the specific concern is. Admittedly, there are different ways in which people have tried to formalize lifelong learning. In our paper, we adopt the LIBERO framework [1], which is well-recognized in lifelong robot learning research. LIBERO presents benchmarks where a robot learns 10 or 20 manipulation tasks sequentially—learning one task after another with limited or no data access to previous and future tasks. This setup exemplifies a lifelong learning scenario by highlighting the challenge of catastrophic forgetting due to data distribution shifts. We have updated **Section 3 PRELIMINARY** at lines 190\~207 to help readers understand our setting from a high-level perspective.
>
> Our approach focuses on mitigating catastrophic forgetting within this established framework. After the lifelong learning stage, we assess the robot's forgetting on scenarios sampled from each task. We introduce weighted local adaptation—emulating the human practice of reviewing forgotten information—to rapidly restore the robot's knowledge of prior tasks using very limited demonstrations. Our results demonstrate that our method effectively reduces catastrophic forgetting in the lifelong robot learning context as defined by LIBERO. We added **Algorithm 1 in Appendix** at lines 993\~1008 to show our pipelines, and  updated our paper to clarify our experiment setup and procedures at **Section 5.1.1** at lines 339\~346 and  **Section 5.1.4** at lines 393\~406.
>
>
> [1] Liu, Bo, et al. "Libero: Benchmarking knowledge transfer for lifelong robot learning." Advances in Neural Information Processing Systems 36 (2024).
>
>
>
>
>
>
>
>
>
>
> > The discussion on unspecified task boundaries in real-world scenarios is also unclear and mischaracterized. Although the paper replaces a one-hot task encoding with an embedding from a vision-language model, this essentially still provides a form of task specification, albeit in a more implicit or 'fuzzy' manner. (Isn’t it true that the agent is aware of the task it’s solving, meaning there’s no true task boundary in the proposed evaluation?)
>
> Thank you for highlighting this concern. We recognize that some concepts were not clearly defined in our initial submission and have updated the paper accordingly. We provide the following explanations to better clarify the definition of task-unawareness of our lifelong robot learning approach.
>
> **Our method is fundamentally task-unaware because it operates without the need for explicit task identifiers or implicit task boundaries** (We have updated our paper in **Section 3 PRELIMINARY** at lines 190\~207). Specifically, our approach applies
>
> - experience replay (ER) during training,
> - demonstration-based retrieval during deployment (for weighted local adaptation), purely based on similarity in visual observations and language descriptions
>
> without any task classification or identification. Our experimental results (**Section 5.2.2 at lines 473\~476**) further show that our weighted local adaptation is robust and fault-tolerant, even with irrelevant demonstrations retrieved.
>
> Determining which task generated a given scenario is also challenging due to overlapping distributions in environmental settings or language descriptions among different tasks. As demonstrated in **Figure 7** (lines 937\~951), even advanced models like BERT and Sentence Similarity cannot reliably distinguish between tasks based on language descriptions. **Figure 8** (lines 972\~992) further shows that our retrieval mechanism is based on demonstration similarity, while it also highlights that substantial environmental and language similarities exist across different tasks.
>
> Regarding the task specifications, the robot does receive some form of task specification necessary for evaluation in LIBERO, this specification is non-trivial and suitable for robots in real lifelong learning scenarios. However, **our notion of "task-unawareness" pertains particularly to our lifelong learning approach**, which, unlike many other lifelong learning algorithms, does not rely on explicit task IDs or task boundaries. For example, dynamic-architecture-based methods like PackNet require task IDs to apply task-specific masks, and regularization-based methods like EWC compute important parameters at task boundaries.
>
> We hope this clarification addresses your concern, and we appreciate your feedback in helping us improve the clarity of our work.

---

> ### Author Response · Authors · 2024-11-25
> **Response to Reviewer z13d (part 2)**
>
> > There is no comparison of the concepts to meta-learning approaches such as [2]. I would argue that their approach resembles having a global network that is fine-tuned to the desired task during testing.
>
> While both methods involve training a global network on multiple tasks and fine-tuning during testing, there are fundamental differences in the learning frameworks and motivations:
> - Meta-learning [1, 2] assumes access to a **full task distribution** during both meta-training and meta-testing phases. The primary objective is to learn a model that can quickly generalize to new tasks within this distribution using few or one-shot learning.
> - Our approach, in contrast, operates in a lifelong learning setting where tasks are learned sequentially. During training on the current task, we have **limited or no access to data from previous or future tasks**, thus with no access to full task distribution during training and testing. Consequently, we must address challenges of catastrophic forgetting due to data distribution shifts, which are central issues in lifelong learning and the focus of our approach.
>
> Therefore, while there are superficial similarities, the fundamental assumptions and goals of our method differ significantly from those of meta-learning approaches. We have updated our paper at **Section 2.4** at lines 157\~161, 185\~188 to highlight this difference.
>
> [1] Finn, C., Abbeel, P., & Levine, S. (2017, July). Model-agnostic meta-learning for fast adaptation of deep networks. In International conference on machine learning (pp. 1126-1135). PMLR.
>
> [2] Finn, C., Yu, T., Zhang, T., Abbeel, P., & Levine, S. (2017, October). One-shot visual imitation learning via meta-learning. In Conference on robot learning (pp. 357-368). PMLR.
>
>
>
> > Would you consider this approach a meta-learning method that aligns with the standard episodic reinforcement learning framework?
>
> As explained in the previous reply, our approach maintains key differences compared to meta-learning approaches.
>
> Regarding standard episodic RL, catastrophic forgetting occurs mainly due to policy changes that lead to data distribution shifts. However, RL agents can often mitigate this by revisiting and relearning from the interactions with the environments.
>
> In contrast, our lifelong robot learning setting involves learning tasks sequentially with limited or no access to data from previous tasks. **This results in irreversible data loss and significant distribution shifts, making catastrophic forgetting a more critical issue.**
>
> Therefore, our approach is distinct in its scenario, problem setting, motivation, framework, etc.—we believe it is not appropriate to categorize it simply as a meta-learning method within the standard episodic RL paradigm.
>
>
>
>
>
>
>
>
> > The description of the experiments and their results is vague and difficult to follow. While I am confident that the authors are skilled and fully understand their work, I encourage them to consider readers like myself, who may be less familiar with it, by providing a more detailed explanation of their setup. For example, what specific tasks are being solved? Are the demonstrations focused on particular tasks? Does each task in the testing phase have a corresponding relevant demonstration?
> While the authors claim lifelong learning is important—a sentiment I share—it is unclear why their chosen evaluation setup suits this purpose. They could provide stronger justification for their design choices.
>
>
> We acknowledge our initial submission wasn’t clear enough and thank you for your suggestions!
>
> We have updated the section **PRELIMINARY** at lines 190\~207 to define the concept of task and scenario, which helps readers understand our setting in a high-level perspective. A more detailed description of our experimental settings is provided in **Section 5.1.4** at lines 382\~406. Additionally, **Algorithm 1 in Appendix** at lines 993\~1008 has been updated to offer a comprehensive overview of the learning and testing procedures.
>
> > Evaluating a lifelong learning agent over only 20 episodes, as done by the authors, seems insufficient and misaligned with the intended concept.
>
> **Concerning the evaluation, we believe there may be a misunderstanding.** We evaluate the robot's performance on each scenario sampled from each task using 20 episodes. Specifically, if a benchmark consists of 10 tasks, we test our model on all of the sampled scenarios, with each scenario evaluated over 20 rollout trials. Additionally, we repeat this process using three different random seeds to ensure robustness. This amounts to a total of 10 tasks × 20 episodes × 3 seeds = 600 evaluation runs for assessing one method in one benchmark.

---

> ### Author Response · Authors · 2024-11-25
> **Response to Reviewer z13d (part 3)**
>
> > The related work section omits many papers that are highly relevant to the context. While the authors do include a few key references, surprisingly, there isn’t a single paper cited from more than five years ago. I'd recommend the authors to look at the following:
>
> Thank you for this valuable suggestion that helps strengthen our paper's scholarly foundation. We have thoroughly enhanced the related work section:
>
> - expanded our discussion to clearly articulate the distinction between lifelong learning and meta-learning at **Section 2.4** at lines 157\~161, 185\~188.
> - enriched **Section 2.1** at lines 105\~113 with foundational papers in lifelong robot learning, including seminal works (Thrun, 1995; Grollman & Jenkins, 2007)
> - the discussion of task-unaware approaches has been broadened in **Section 2.2** at lines 139\~145, incorporating highly relevant works  (Aljundi et al., 2019a; Aljundi et al., 2018).
>
>
>
>
> > How would this approach perform if only partial task demonstrations were available? Could it potentially combine these fragments to form a complete solution?
>
>
> Thank you for this insightful question about partial demonstrations! Your question touches on a fascinating aspect of real-world robotic learning where complete demonstrations may not always be available.
>
> We argue that our retrieval-based weighted local adaptation system shows strong potential for handling partial demonstrations through its flexible architecture and robustness:
>
> The dual-embedding retrieval system leverages both initial observation embeddings and task description embeddings. The weighting between these components can be dynamically adjusted through the parameters in **Equation 2** at lines 254\~255.
>
> When working with partial demonstrations, we can increase the weight of description-based retrieval. This enables the system to identify relevant fragments that share similar task objectives. This could enable the integration of multiple partial demonstrations with semantic consistency to reconstruct a complete solution.
>
> We look forward to exploring this direction further in future work, as it represents an important practical consideration for real-world deployment.
>
>
> Let us know if you have any further questions. Look forward to your reply!

---

> ### Author Response · Authors · 2024-12-01
> **Reply to Reviewer z13d**
>
> Dear reviewer z13d,
>
> Thank you again for your constructive comments! As we are near the end of the discussion period, we hope to know whether our responses have addressed your concerns. We look forward to your further feedback and suggestions.
>
> Best regards,
>
> Authors of 12126

---

> ### Author Response · Authors · 2024-12-03
> **Reminder for the response**
>
> Dear Reviewer z13d,
>
> As the discussion period is nearing its end, we would like to confirm whether our replies and the updates to the submission have addressed your concerns. **If you have any further comments, we are happy to reply to them before the author response deadline on December 3 (AOE)**.
>
> Thank you again for your insightful suggestions!
>
> Best regards,
> Authors of Paper 12126

---

### Author Response · Authors · 2024-11-27
**General Response to All Reviewers and AC**

We sincerely thank all reviewers for their detailed feedback and constructive suggestions. Your insights have been instrumental in refining our work. Below, we summarize the major revisions (highlighted with yellow) in the paper based on your comments:

1. Related Work (Sec. 2, Reviewers z13d, qCFh):

- Expanded with additional relevant papers.
- Highlighted distinctions from prior work, particularly meta-learning approaches.
2. Preliminaries (Sec. 3, Reviewers z13d, UrfP, qCFh):

- Improved clarity on task-unawareness and the lifelong robot learning pipeline.
3. Few explanations in Experimental Setup:

- Added explanations for high variance in results. (Sec. 5.1.3, Reviewer mMan)
- Clarified experimental settings. (Sec. 5.1.4, Reviewers z13d, qCFh)
4. Appendix Updates:
- Added figures (7–9) to illustrate blurry task boundary effect, and data retrieval and selective weighting mechanisms. (Reviewer z13d, UrfP, qCFh)
- Included Algorithm 1 (pseudocode) for a clear pipeline overview. (Reviewers z13d, qCFh)
- Enhanced justification of selective forgetting (Sec. A.4) and tested hyperparameter sensitivity in Figure 10. (Reviewers UrfP, qCFh)
- Added a discussion (Sec. A.6) about the solution to potential forgetting during local adaptation. (Reviewer UrfP)

**These updates address key suggestions without altering our motivation, approach, implementation, experimental settings, or results**.

We welcome further concerns or questions and look forward to your feedback! Thank you again for your invaluable input!

---

### Meta-Review · Area_Chair_8cUd · 2024-12-20

**Metareview:**

This paper introduces a method for retrieval-based fast adaptation in a lifelong learning setting. The proposed approach leverages retrieved prior data to augment adaptation, using both visual similarity and textual embedding similarity of task descriptions. Experiments conducted on a simulated benchmark demonstrate the effectiveness of the method compared to existing lifelong learning techniques.

The reviewers acknowledged the novelty of the proposed method and appreciate its conceptual simplicity. However, significant concerns were raised regarding the clarity of the paper’s presentation and the lack of sufficient baselines in the experimental evaluation. While the authors addressed some of the clarity issues during the discussion period, the experimental results remain unconvincing to merit acceptance.

Although reviewer qCFh provided a positive recommendation, the consensus among the other reviewers is that this work can be further improved.

**Additional Comments On Reviewer Discussion:**

The authors clarified on implementation details during discussion and added additional discussion of related works to the revised version of the paper.

---

### Decision · Program_Chairs · 2025-01-22

Reject